



# The Arctic Low-Level Mixed-Phase Haze Regime and its Microphysical Differences to Mixed-Phase Clouds

Manuel Moser[1], Christiane Voigt[1,2], Oliver Eppers[3], Johannes Lucke[1,4,10], Elena De La Torre Castro[1], Johanna Mayer[1], Regis Dupuy[5], Guillaume Mioche[5], Olivier Jourdan[5], Hans-Christian Clemen[3], Johannes Schneider[3], Philipp Joppe[2,3], Stephan Mertes[6], Bruno Wetzel[6], Stephan Borrmann[2,3], Marcus Klingebiel[7], Mario Mech[8], Christof Lüpkes[9], Susanne Crewell[8], André Ehrlich[7], Andreas Herber[9], and Manfred Wendisch[7]

[1]Institut für Physik der Atmosphäre, Deutsches Zentrum für Luft- und Raumfahrt, Weßling, Germany
[2]Institut für Physik der Atmosphäre, Johannes Gutenberg-Universität, Mainz, Germany
[3]Max Planck Institute for Chemistry, Mainz, Germany
[4]Faculty of Aerospace Engineering, Delft University of Technology, Delft, Netherlands
[5]Laboratoire de Météorologie Physique, Université Clermont Auvergne, Clermont-Ferrand, France
[6]Leibniz–Institut für Troposphärenforschung, Leipzig, Germany
[7]Leipziger Institut für Meteorologie, Universität Leipzig, Leipzig, Germany
[8]Institut für Geophysik und Meteorologie, Universität zu Köln, Cologne, Germany
[9]Alfred–Wegener–Institut, Helmholtz–Zentrum für Polar– und Meeresforschung, Bremerhaven, Germany
[10]Rail Tec Arsenal, Vienna 1210, Austria

**Correspondence:** Manuel Moser (manuel.moser@dlr.de)

**Abstract.** A comprehensive in-situ dataset of low-level Arctic clouds was collected in the Fram Strait during the HALO-(AC)³ campaign in spring 2022 using the research aircraft Polar 6. The clouds observed at altitudes below 1000 m were frequently in a mixed-phase state. We demonstrate that despite comparable optical properties, classic mixed-phase clouds (MPC) and mixed-phase haze (MPH) can be distinguished on the basis of their microphysical properties. While the thermodynamic phases

5 of the particles within the MPH are similar to those in the MPC, the supercooled droplets observed in MPC are replaced by large (> 3 µm) wet aerosol particles in MPH. Furthermore, the particle number concentration measured in MPH is reduced by approximately 3 orders of magnitude compared to MPC. MPH is observed in subsaturated air with respect to water, suggesting that the small liquid particles are haze droplets and are in equilibrium below the activation threshold to form cloud droplets. Chemical analysis suggested that the haze particles contained significant amounts of sea salt. Additional in-situ measurements

10 with an optical particle counter indicated that their number concentration was two times larger over the sea ice compared to the open ocean. Furthermore, measurements of the vertical distribution of the thermodynamic phases in low-level Arctic clouds revealed a characteristic structure, with a liquid regime frequently occurring at the top of the atmospheric boundary layer, followed by MPCs, and an MPH layer below.

The findings from this study enhance our understanding of the microphysical composition of clouds in mixed-phase condi-

15 tions.



# 1 Introduction

Over the past few decades, the Arctic region has drastically changed in response to global warming (Jeffries et al., 2013; IPCC, 2021). The accelerated warming observed at high latitudes, with a rate that is more than twice as fast as the global average (Overland et al., 2019), is known as Arctic amplification (Serreze and Francis, 2006; Wendisch et al., 2023a). Projections show that even under current efforts to mitigate greenhouse gas emissions, the Arctic will be transformed beyond contemporary recognition (Stroeve et al., 2025). Many factors influencing Arctic amplification are discussed (Wendisch et al., 2023a), including the reduction of albedo due to the decrease of the mean sea ice extent (Budyko, 1969; Sellers, 1969; Druckenmiller et al., 2022), the change in lapse rate (Pithan and Mauritsen, 2014), and changes of large scale weather patterns (Francis and Vavrus, 2012; Mann et al., 2017; Coumou et al., 2018; Kretschmer et al., 2018; Heukamp et al., 2023). The amplification is likely to be driven by a combination of multiple factors. Clouds may play a key role in the processes underlying the intense mean temperature rise at high latitudes (Wendisch et al., 2019). Low-level clouds in the Arctic are often found in a mixed-phase state (Shupe et al., 2006; Morrison et al., 2011), representing a three-phase non-equilibrium system consisting of water vapor, ice particles, and coexisting supercooled liquid water droplets. These cloud systems can persist in a quasi-steady state due to continuous water supply driven by local dynamical processes (Morrison et al., 2011). Together with stratiform liquid water clouds, Arctic mixed-phase clouds are important contributors to the Arctic surface radiation budget (Shupe and Intrieri, 2004; Wendisch et al., 2023b). In particular, the size, shape, and thermodynamic phase of the cloud particles influence the atmospheric energy fluxes and are often poorly represented in observations and models (Naud et al., 2014; Bodas-Salcedo et al., 2016; McCoy et al., 2016; Tan and Storelvmo, 2019; Wendisch et al., 2019; Kretzschmar et al., 2020; Marsing et al., 2023), contributing to the limited confidence in quantifying the cloud feedback in the Arctic climate system (Morrison et al., 2011; Bock et al., 2020; IPCC, 2021).

Several studies have investigated Arctic mixed-phase clouds using in-situ measurements, including those by McFarquhar et al. (2007), Lawson and Zuidema (2009), Klingebiel et al. (2015), Young et al. (2016), Mioche et al. (2017), Järvinen et al. (2023) and Moser et al. (2023b). These studies analyzed the microphysical processes of mixed-phase clouds and their vertical structure, examined their persistence and the influence of meteorological conditions on cloud properties, and suggested microphysical cloud parameterizations for models. However, most low-level in-situ cloud studies apply specific cloud thresholds, e.g. in liquid water content, or number concentration, that may have excluded optically thin clouds. For instance, Kirschler et al. (2022), Kirschler et al. (2023), and Sorooshian et al. (2023) defined cloud periods based on cloud water content exceeding $0.01 - 0.02\,\mathrm{g\,m^{-3}}$ and a particle number concentration greater than $10 - 20\,\mathrm{cm^{-3}}$. Similarly, Young et al. (2016) distinguished in-cloud and out-of-cloud observations using a threshold of $0.01\,\mathrm{g\,m^{-3}}$. Such thresholds may lead to the omission of thin clouds, despite their potential importance in the atmospheric radiation budget. Wendisch et al. (2013) show that the radiative forcing of low-level clouds with optical depths below 2 are particularly sensitive to small changes in optical thickness. Costa et al. (2017) observed a low number concentration ($N < 1\,\mathrm{cm^{-3}}$) in some cloud types when analysing a large in-situ data set of clouds in the mixed-phase temperature range. They have hypothesised that these clouds may have been formed by the drying of mixed-phase clouds via the Wegener-Bergeron-Findeisen process.



In this study, we conduct a detailed investigation of this cloud regime, which we call mixed-phase haze, characterized by relatively low particle number concentrations, analyzing its microphysical properties and differentiating it from classic mixed-phase clouds.

The article is structured as follows. Section 2 introduces the HALO-(AC)[3] field campaign, describes the instrumentation for the in-situ measurements as well as the data processing methods, and presents the prevailing environmental and meteorological conditions together with complementary dropsonde data. Section 3 introduces the mixed-phase haze cloud regime and compares its microphysical properties with those of classic mixed-phase clouds, analyzes the chemical composition and particle number concentration of the haze droplets, and investigates the vertical distribution of cloud thermodynamic phases in the Arctic atmospheric boundary layer. Section 4 summarizes the main findings and discusses their implications.

## 2 Methods

### 2.1 The aircraft field campaign HALO-(AC)[3]

The HALO-(AC)[3] field campaign, conducted in March and April 2022, was a comprehensive effort to observe and analyze air mass transformations during warm-air intrusions (WAIs) and cold-air outbreaks (CAOs) over the Norwegian and Greenland Seas, the Fram Strait, and the central Arctic Ocean (Wendisch et al., 2025). The campaign utilized three research aircraft: The High Altitude and Long Range Research Aircraft (HALO; Krautstrunk and Giez, 2012; Stevens et al., 2019), operated by the German Aerospace Center (DLR), and the Polar 5 and Polar 6 aircraft (Wesche et al., 2016), operated by the German Alfred Wegener Institute. At the same time, similar flight experiments were conducted in partially close proximity by the British Facility for Airborne Atmospheric Measurements (FAAM) and the French Avions de Transport Régional (ATR) aircraft. All aircraft operated in coordination, covering a broad range of altitudes and spatial scales. An overview and detailed description of the HALO-(AC)[3] field campaign is provided by Wendisch et al. (2024). A comprehensive description of the data collected on board the aircraft can be found in Ehrlich et al. (2025).

This study focuses on the data collected by the Polar 6 aircraft. It is a modern version of the former Douglas DC-3, which had been modified by Basler Turbo Conversions for operations in extreme polar conditions (BT-67; Wesche et al., 2016). It was equipped with in-situ instrumentation to analyze cloud and aerosol particles. The research flights were conducted from Longyearbyen (LYR; 78° N, 15° E) in the region of the Fram Strait between Greenland and Svalbard. The flight strategy followed a similar approach as applied during the previous campaigns AFLUX and MOSAiC-ACA (Mech et al., 2022a), focusing on in-situ cloud measurements in the atmospheric boundary layer over the sea ice and the open ocean. The Polar 6 flight plans included horizontal flight legs at different altitudes through clouds, as well as vertical profile measurements involving climbs or descents along a straight trajectory through cloud layers (Mech et al., 2022a). During HALO-(AC)[3], Polar 6 was temporally and spatially coordinated with HALO and Polar 5 to enable collocated in-situ and remote sensing observations, expanding on similar approaches employed during the ACLOUD field campaign (Ehrlich et al., 2019; Wendisch et al., 2019).

The region where the in-situ measurements were conducted is shown in Fig. 1. The figure also displays the sea ice concentration (SIC) from satellite observations by the Advanced Microwave Scanning Radiometer 2 (AMSR2) instrument (Spreen et al.,



2008), at a representative time for the campaign (30 March 2022). The microphysical low-level cloud dataset from Polar 6 consists of a total of 19.4 h ($< 1000$ m, and cloud threshold CWC $> 2 \times 10^{-4}$ g m$^{-3}$ according to Moser et al., 2023b), collected during 13 flights in March and April 2022.

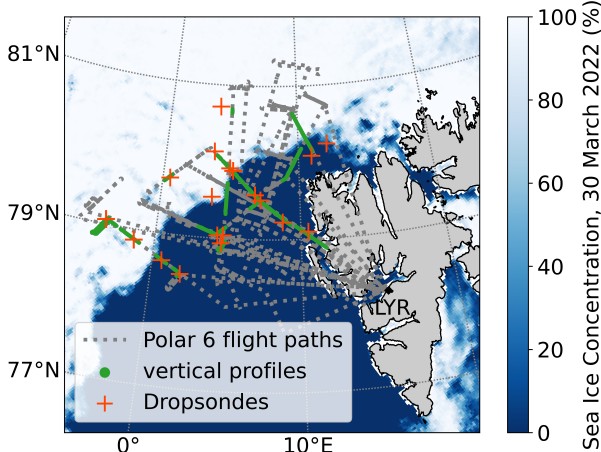

**Figure 1.** The map shows the region where in-situ cloud measurements were conducted with the Polar 6 aircraft. The green markers indicate the positions of vertical profiles. Orange crosses represent the locations of dropsondes collocated with the in-situ measurements used in this study. The background displays the sea ice concentration at the midpoint (30 March 2022) of the campaign, as recorded by the Advanced Microwave Scanning Radiometer 2 (AMSR2).

## 2.2 Cloud and aerosol in-situ instrumentation and the data processing

The Polar 6 aircraft was equipped with an advanced payload for in-situ instruments to detect cloud particles across the size range typically found in Arctic low-level clouds. For the smallest particles, between 3 µm and 50 µm, a Cloud Droplet Probe (CDP; Lance et al., 2010) scattering instrument was used to determine the particle sizes. Larger particles were measured using optical array probes, specifically the Cloud Imaging Probe (CIP; Baumgardner et al., 2001) for particle diameters between 15 µm and 960 µm, and the Precipitation Imaging Probe (PIP; Baumgardner et al., 2001) for particle diameters ranging from 100 µm to 6.4 mm. By combining the data from these three instruments, a size spectrum of cloud particles, covering a range from 3 µm to 6.4 mm was established. The data processing to derive microphysical parameters such as $N$, effective diameter ($D_{\text{eff}}$), cloud water content (CWC), liquid water content (LWC), and ice water content (IWC) followed the same methodology as described by Moser et al. (2023b) and Mech et al. (2022a). In addition, aerosol instruments complemented particle measurements below the detectable size range of the cloud probes. An optical particle counter (Grimm Sky-OPC, model 1.129; Heim et al., 2008; Bundke et al., 2015) measured the size distribution and concentration of aerosol particles in the size range between 0.25 and 40 µm by analyzing the scattered light as particles pass through a laser beam. However,



the upper size limit of the OPC was reduced to approximately 6 μm during HALO-(AC)[3] due to the transmission efficiencies of the inlet system and the sampling lines to the instrument. Furthermore, the single particle mass spectrometer ALABAMA (Aircraft-based Laser ABlation Aerosol MAss spectrometer; Brands et al., 2011; Köllner et al., 2017; Clemen et al., 2020) analyzed the chemical composition of individual aerosol particles in real time using laser-induced desorption/ ionization mass spectrometry. The size range of the ALABAMA ranged between 230 nm and 3 μm with 50 % detection efficiency (Clemen et al., 2020). The recorded mass spectra from the measurements conducted during the HALO-(AC)[3] flights were processed and classified into 37 clusters referring to different particle types (Ehrlich et al., 2025). Clusters containing sodium chloride (NaCl) signals were merged together to a 'sea spray' particle type. Besides the chemical composition, also the sizes of the individual particles are measured by the ALABAMA using the scattering signal at two consecutive detection laser stages. Both instruments, the OPC and the ALABAMA, were mostly operated behind the Counterflow Virtual Impactor (CVI) inlet (Ehrlich et al., 2025). The CVI enabled sampling of cloud particle residues, which remain after evaporation or sublimation of cloud droplets or ice crystals, respectively, by separating cloud particles from aerosol particles using a counterflow within the inlet system. Outside clouds, the CVI was used as particle and trace gas inlet by switching off the counterflow. In this study, only aerosol measurements outside of clouds were considered. The transmission efficiency of the inlet lines to the ALABAMA and the OPC was near unity for particles from 50 nm to about 1 μm and reduced to 80 % at 2 μm, 10 % at 4 μm and 0 % at 6 μm.

In this paper, two methodological approaches are applied based on different types of flight patterns to investigate mixed-phase conditions. First, in Sect. 3.1 we analyze and compare the microphysical properties of mixed-phase haze and mixed-phase clouds, as well as the chemical composition of the aerosol particles, with high statistical accuracy. For this purpose, only the in-situ data from the horizontal flight legs are used. Second, the thermodynamic phases are investigated in relation to environmental conditions (Sect. 3.1.2) and normalized altitude (Sect. 3.2). In this analysis, vertical flight legs, i.e., straight ascents and descents through the cloud layer, are used to characterize the vertical distribution of cloud properties. This approach is applied to correlate in-situ aircraft data with dropsonde measurements. The results from measurements conducted on vertical flight legs are statistically less significant than those from horizontal legs, however, phase determination at 1 Hz frequency, can be applied here. Additionally, during ascent and descent, different airflow conditions may affect the isocinetic sampling of the instruments, potentially leading to an increased measurement uncertainty (Moser et al., 2023b). However, previous studies have demonstrated that vertical analyses with in-situ cloud measurements are feasible (Mioche et al., 2017; Taylor et al., 2019; Schima et al., 2022; Järvinen et al., 2023; Braga et al., 2025).

By Moser et al. (2023b), a method was introduced for determining the thermodynamic phase in low-level Arctic clouds based on microphysical characteristics. The approach involves plotting 1 Hz cloud data in $N$ - $D_{\mathrm{eff}}$ space, where visible clusters can be linked to specific cloud regimes including liquid, aerosol, mixed-phase, or ice regimes. This method was validated using the Polar Nephelometer and is applied here. However, in the spring data from Moser et al. (2023b), cloud particles smaller than 50 μm were measured with a Cloud Aerosol Spectrometer (CAS), which is slightly more sensitive to small particles (Braga et al., 2017). As a result, determining the thresholds for the individual regimes using the same procedure as in Moser et al. (2023b) leads to slightly different thresholds of the regimes. In this study, the thresholds of the individual regimes are extended to include at least 80 % of the data ($D_{\mathrm{eff}}$ and $N$ values between the 10th and 90th percentiles) from the respective regime data



peak in the $D_{eff}$ - $N$ space from the HALO-(AC)[3] in-situ dataset. Although these thresholds are not substantially different from

those presented in Moser et al. (2023b), they are necessary to optimize the cloud dataset assigned to a specific thermodynamic cloud phase. Table 1 lists the upper and lower limits of $N$ and $D_{eff}$ that define the respective regimes. When a data point falls within a specific regime, the cloud thermodynamic phase is derived as follows: Regimes 1a and 1b correspond to ice particles, regimes 2a and 2b, as we show in Sect. 3.1, to mixed-phase haze particles, regime 2c to the classic mixture of liquid water and ice particles, regime 3 to liquid water particles, and regime 4 is associated with aerosols.

**Table 1.** Classification of different cloud regimes based on the in-situ measured $N$ and $D_{eff}$.

| Regime | Thermodynamic phase | Lower limit $N$ | Upper limit $N$ | Lower limit $D_{eff}$ | Upper limit $D_{eff}$ |
|---|---|---|---|---|---|
| 1a | Ice | $10\,\mathrm{m}^{-3}$ | $196\,\mathrm{m}^{-3}$ | 0.4 mm | 2.7 mm |
| 1b | Ice | $513\,\mathrm{m}^{-3}$ | $1.9{\times}10^{4}\,\mathrm{m}^{-3}$ | 0.34 mm | 2.4 mm |
| 2a | MPH | $6.1{\times}10^{4}\,\mathrm{m}^{-3}$ | $7.1{\times}10^{5}\,\mathrm{m}^{-3}$ | 0.15 mm | 0.82 mm |
| 2b | MPH | $6.0{\times}10^{4}\,\mathrm{m}^{-3}$ | $4.3{\times}10^{5}\,\mathrm{m}^{-3}$ | 1.1 mm | 3.6 mm |
| 2c | MPC | $1.8{\times}10^{7}\,\mathrm{m}^{-3}$ | $2.1{\times}10^{8}\,\mathrm{m}^{-3}$ | 0.07 mm | 0.99 mm |
| 3 | Liquid | $2.4{\times}10^{7}\,\mathrm{m}^{-3}$ | $2.5{\times}10^{8}\,\mathrm{m}^{-3}$ | 9 $\mu$m | 38 $\mu$m |
| 4 | Aerosol | $6.0{\times}10^{4}\,\mathrm{m}^{-3}$ | $4.0{\times}10^{5}\,\mathrm{m}^{-3}$ | 3 $\mu$m | 10 $\mu$m |

**2.3 Environmental and meteorological conditions**

The meteorological conditions in the Fram Strait during the HALO-(AC)[3] in-situ flights of Polar 6 were predominantly characterized by CAOs. Exceptions include 20 March and 10 April, where the prevailing situations were a WAI, and 8 April, when the meteorological condition was dominated by a polar low (Walbröl et al., 2024; Wendisch et al., 2024). Backward trajectory analyses were conducted to determine the dominant surface type over which the low-level air masses had resided during the

24 h prior to their in-situ measurement by Polar 6. This analysis followed the same methodology as described in Moser et al. (2023b), employing the Hybrid Single-Particle Lagrangian Integrated Trajectory model (HYSPLIT) (Stein et al., 2015; Rolph et al., 2017) with the Global Forecast System (GFS) at a 0.25° horizontal resolution as meteorological input.

  To classify the ocean surface on a daily resolution, remote sensing data from the Global Change Observation Mission-Water (GCOM-W1) satellite were used. The AMSR2 instrument aboard the satellite recorded SIC at a spatial resolution of 3.125

km (Spreen et al., 2008). In this study, SIC values greater than 80 % are classified as sea ice, SIC values below 20 % as open ocean, and values between 20 % and 80 % as the marginal sea ice zone (MIZ). The SIC and the derived air mass origins for the vertical flight legs are listed in Table A1. The classification "ocean/land" indicates that the low-level air mass was mainly influenced by the open ocean within the past 24 h. However, in some specific cases, an influence from the Svalbard land region cannot be ruled out.

To analyze the environmental conditions of the atmosphere, profiles of temperature ($T$) and relative humidity with respect to water (RH$_\mathrm{w}$) measured by dropsondes were used (George et al., 2024). To calculate the relative humidity with respect to ice (RH$_\mathrm{ice}$), the equilibrium vapor pressure definitions provided by World Meteorological Organization (2012) are used. While





Polar 6 conducted in-situ measurements of low-level clouds, HALO and Polar 5 flew above, releasing dropsondes into the flight path of Polar 6 (Ehrlich et al., 2025). For each vertical flight profile of Polar 6, the nearest dropsonde in time and space, either

launched from Polar 6 or HALO, was selected. The criteria for accepting a dropsonde to correlate with in-situ profiles were a distance of less than 70 km and a time difference of less than 2 h. In total, 33 vertical flight legs with sufficient in-situ data within and just above the atmospheric boundary layer (ABL) were flown by Polar 6. These were matched with dropsondes based on the defined criteria. The start and end times of each vertical profile from Polar 6, along with the corresponding dropsonde ID, are presented in Table A1. Meteorological data from the nose boom on Polar 6 were not used in this study due to icing

conditions and associated de-icing issues, which limited the availability of reliable data to only a few days. However, these data were used to validate the correlation approach between Polar 6 and the dropsonde data. All temperature profiles from the dropsondes used in this study are shown in gray in Fig. 2. Additionally the dropsondes were classified based on satellite data to determine whether they were deployed over sea ice (SIC > 80 %), the open ocean (SIC < 20 %), or the MIZ (20 % ≤ SIC ≤ 80 %). From the measured temperature profiles of the dropsondes, the ABL height for each vertical flight leg is estimated. The

ABL height is identified as the lowest altitude, at which a minimum in the temperature occurs. Additionally to the ABL height, the measured temperature at this height is provided in Table A1.

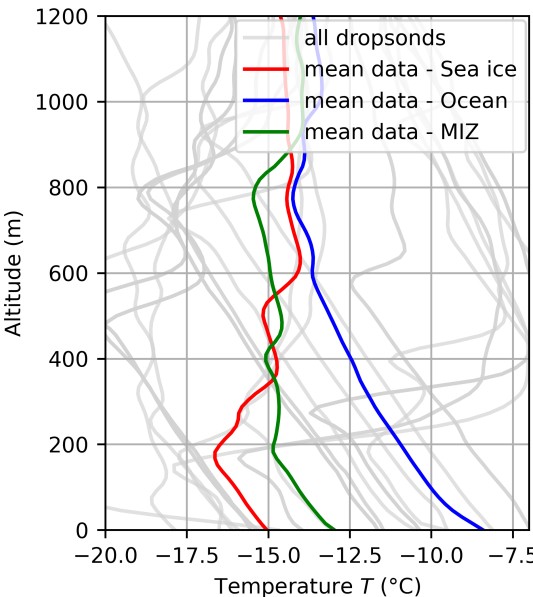

**Figure 2.** Temperature profiles measured by dropsondes from HALO and Polar 5. The gray lines represent all dropsondes used in the analysis in Sect. 3.1.2 and 3.2. The red line shows the averaged temperature profile from dropsondes released over sea ice, the blue line represents the averaged temperature profiles over the open ocean, and the green line corresponds to the averaged temperature profiles in the marginal sea ice zone (MIZ). All dropsondes used in this figure are listed in Table A1.

As shown in Fig. 2, the ABL height strongly depends on the underlying surface. This figure presents temperature profiles from all utilized dropsondes as a function of altitude, with mean temperature profiles over sea ice, the MIZ, and the open



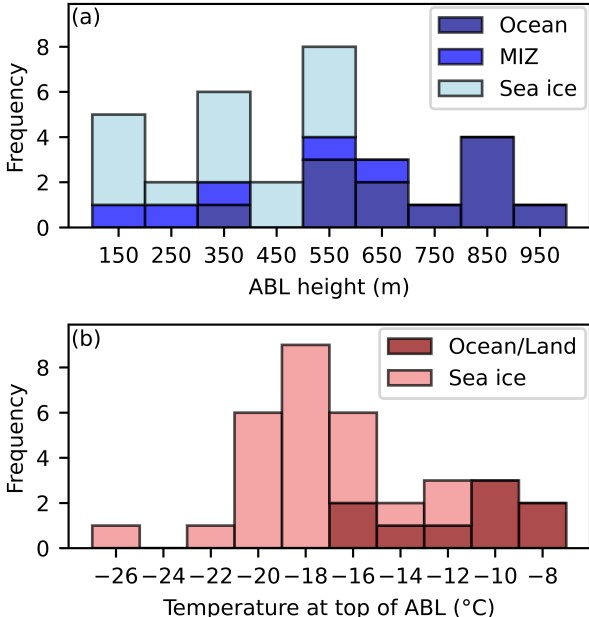

**Figure 3.** Data extracted from the dropsonde profiles: (a) shows the distribution of boundary layer top altitudes, color-coded by underlying surface type: Ocean, marginal ice zone, and sea ice, based on the mean sea ice concentration. (b) presents the distribution of temperatures at the top of the boundary layer, distinguished by air mass origin: 'Ocean/Land' and 'Sea ice' based on backward trajectory analyses.

ocean. The measured ABL heights and the temperatures at the ABL heights are shown as histograms in Fig. 3. While the ABL

is relatively shallow over sea ice and within the MIZ, it extends to significantly higher altitudes over the ocean when convection sets in. All ABL heights below 505 m, with only one exception, were measured over sea ice or within the MIZ. Conversely, ABL heights above 630 m were exclusively observed over the ocean. Consequently, the left part (< 500 m) of the distribution in Fig. 3 (a) represents ABL heights over sea ice and over the MIZ, while the right part (> 600 m) corresponds to ABL heights over the open ocean. The central peak in the histogram (at 550 m) contains ABL heights measured across all surface types.

Figure 3 (b) displays the distribution of temperatures measured at the ABL top. The observed temperatures range from -22°C to -8°C, with a peak around -18°C. Based on backward trajectory analyses, all temperatures above -12°C are associated with air masses that had interacted with either the ocean or land within the past 24 h. Conversely, for temperatures below -16.4°C, the corresponding air masses originated from sea ice. Very similar temperature variations at the ABL top relative to air mass origin were previously reported by Mioche et al. (2017). The analysis of Fig. 3 indicates that the vertical structure of the ABL, as

sampled by Polar 6, is shaped by both the underlying surface conditions and the air mass origin. While the ABL height shows a dependence on the surface type (i.e., higher over open ocean, lower over sea ice), the temperature at the ABL top appears to be influenced by the air mass origin. This suggests that the synoptic situation primarily determines the temperature at the top of the ABL during HALO-(AC)[3], whereas local surface-driven processes control its vertical extent.





To consider the variable ABL height, the analyzed vertical profiles were normalized by altitude. The observed altitude, $z$, is normalized by the boundary layer height $z_{\mathrm{BL}}$. This results in a dimensionless normalized altitude, $z_{\mathrm{norm}}$, calculated as:

$$z_{\mathrm{norm}} = \frac{z}{z_{\mathrm{BL}}}, \tag{1}$$

where $z_{\mathrm{norm}} = 1$ represents the top of the boundary layer, while $z_{\mathrm{norm}} = 0$ corresponds to the surface. Values greater than 1 indicate altitudes above the boundary layer. Note that the normalized altitude applied here differs from earlier mixed-phase studies that scaled by individual cloud depth only (Mioche et al., 2017; Järvinen et al., 2023; Chechin et al., 2023).

## 3    Results and discussion

### 3.1    Microphysical properties of mixed-phase clouds and mixed-phase haze

In the following, the differences in microphysical properties between classic mixed-phase clouds and mixed-phase haze are analyzed, focusing on particle size distributions, environmental conditions, and the aerosols within the mixed-phase haze.

### 3.1.1    Particle size distribution

We analyze mixed-phase regimes (2a, 2b, 2c) by applying the phase-detection algorithm of Moser et al. (2023b) to the HALO-(AC)[3] in-situ data, focusing on their particle size distributions and microphysical composition. Regime 2c corresponds to a mixed-phase cloud typically described in literature (McFarquhar et al., 2007; Korolev et al., 2017; Morrison et al., 2011) and now will be referred to a classic mixed-phase cloud (MPC). However, in the following it is shown that the mixed-phase states, as they occur in regimes 2a and 2b, differ from a classic mixed-phase cloud, and that the term Arctic low-level mixed-phase haze or mixed-phase haze (MPH) adequately describes the regime 2a and 2b. As shown by Moser et al. (2023b) with the Polar Nephelometer, all three regimes show optical properties typical for a classic mixed-phase cloud. In terms of microphysics, however, regimes 2a and 2b differ substantially from regime 2c. The microphysical differences between regimes 2a and 2b are minor and mainly related to surface conditions (Moser et al., 2023b), while both show pronounced differences compared to regime 2c. Therefore, for the subsequent analysis, it is reasonable to combine regimes 2a and 2b into a single category representing the MPH regime, to enable a direct comparison with regime 2c (MPC).

Figure 4 presents particle size distributions (PSDs) measured under MPC (blue) and MPH (green) conditions. PSDs were calculated using data collected during horizontal flight legs below 1000 m using the combined particle measurement system. The solid lines represent the median PSDs, while the shaded areas indicate the 25th and 75th percentiles, derived from 5000 PSDs generated through bootstrapping. Each PSD was constructed from three randomly selected 1 Hz data points corresponding to the respective cloud regime. For particles larger than 50 μm, the PSD for MPC and MPH are quite similar, indicating that the size distribution of ice crystals in both regimes is comparable, as ice crystals are expected in this size range. The most significant difference between the two PSDs is observed for particles smaller than 50 μm. In classic mixed-phase clouds, the PSD shows a clear size mode between 3 μm and 30 μm caused by supercooled liquid water droplets, with a maximum at approximately 10 μm. This droplet mode is absent in the PSD of the MPH regime. Instead, the median PSD increases from





10 μm with decreasing size with a maximum concentration at 3 μm, which is at the lower detection limit of the CDP. Most

particles in this size range are measured between 3 μm and 6 μm. The number concentration of the small particles is reduced

by more than a factor of 1000 in MPH compared to MPC. It is hypothesized that, while the local maximum in the PSD of

classic mixed-phase clouds (2c) is caused by supercooled liquid water droplets, the particles dominating the particle number

concentration in the mixed-phase haze regimes (2a and 2b) consists of haze droplets (Hobbs and Wallace, 2006) which might

be dissolved sea salt particles. Consequently, we conclude that the mixed-phase haze regimes 2a and 2b consist of a mixture

of small (< 6 μm) wet sea salt aerosols (SSA) and larger ice crystals. The gap in Fig. 4 between the larger ice crystals and

the droplets or haze droplets arises from the low number concentration of cloud particles in this intermediate size range. The

individual PSDs are computed at a frequency of 1 Hz, which is insufficient to accumulate a sufficiently large statistical sample

to resolve this size range.

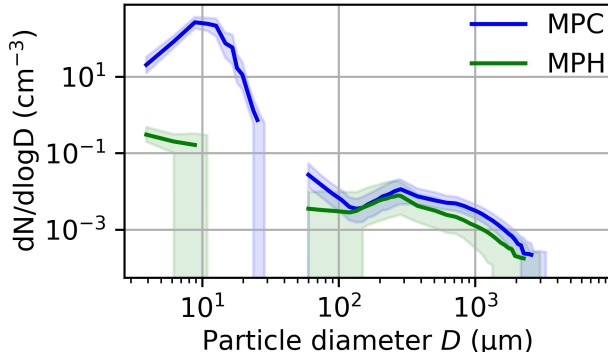

**Figure 4.** Particle size distribution of classic mixed-phase clouds (blue) and mixed-phase haze (green). Solid lines give a median value and
the shaded areas represent the respective variability, calculated by the 25th and 75th percentiles.

230       The microphysical properties, including $N$, $D_{\text{eff}}$, LWC, IWC, and CWC, are summarized in Table 2 for both MPC and MPH.

The microphysical values are presented for the entire PSD in the top row, followed by those for particles smaller than 50 μm

(assumed to represent liquid water or haze droplets) in the middle row, and for particles larger than 50 μm (assumed to be ice

particles) in the lowest row. This threshold to distinguish between liquid droplets and ice crystals is appropriate for the majority

of low-level Arctic clouds (McFarquhar et al., 2007; Korolev et al., 2017). MPCs are characterized by a significantly higher

median particle number concentration ($\tilde{N}$) of 104.6 cm$^{-3}$, compared to only 0.1 cm$^{-3}$ in MPH. This increased total number

concentration in MPCs is dominated by supercooled droplets, while in MPH, $\tilde{N}$ is dominated by wet SSA, as indicated by

the subsequent chemical analysis. The number concentration of ice particles is about 3 orders of magnitude smaller than the

total number concentration in MPCs and about a factor of 50 smaller in MPH, making their contribution to the total number

concentration negligible. The median effective diameter ($\tilde{D}_{\text{eff}}$) calculated for the entire PSD is larger in MPH with 685 μm

compared to MPC with 211 μm. This difference is attributed to the absence of the size mode corresponding to the supercooled

droplets in MPH. The absence of this mode also leads to a slightly higher $\tilde{D}_{\text{eff}}$ for particles smaller than 50 μm in the MPH

regime. However, the $\tilde{D}_{\text{eff}}$ for particles larger than 50 μm indicates that ice crystals in MPCs are larger than those in MPH



though particles > 100 μm appear to be equally represented. The CWC is calculated as the sum of liquid water content (LWC) and ice water content (IWC) (for details, see Moser et al., 2023b). The median CWC in MPCs is approximately 20 times higher

than in MPH. While the CWC in MPCs is fairly evenly distributed between liquid water and ice particles, in MPH, over 99 % of the cloud water content is associated with the ice phase.

**Table 2.** Microphysical properties of low-level mixed-phase cloud regimes and low-level Arctic mixed-phase haze regimes: Median number concentration $\tilde{N}$, median effective diameter $\tilde{D}_{\mathrm{eff}}$ and median cloud water content $\widetilde{\mathrm{CWC}}$. The values in the square brackets give the 25th and 75th percentile, respectively. The microphysical properties are calculated from all detected cloud particles, as well as for particles smaller than 50 μm (assumed to be liquid water for mixed-phase clouds and assumed to be wet aerosols for mixed-phase haze) and for particles larger than 50 μm (assumed to be ice). The data consists of in-situ measurements during horizontal flight legs only and at altitudes below 1000 m.

| | Mixed-phase cloud (MPC) | Mixed-phase haze (MPH) |
|---|---|---|
| $\tilde{N}$ (cm$^{-3}$) | 104.63 [57.74 / 139.24] | 0.13 [0.07 / 0.25] |
| $\tilde{D}_{\mathrm{eff}}$ (μm) | 211 [121 / 400] | 685 [392 / 1445] |
| $\widetilde{\mathrm{CWC}}$ (g m$^{-3}$) | 0.17 [0.09 / 0.26] | (21.91 [3.87 / 244.20])×10$^{-3}$ |
| $\tilde{N}_{<50\mu m}$ (cm$^{-3}$) | 104.62 [57.73 / 139.23] | 0.13 [0.07 / 0.24] |
| $\tilde{D}_{\mathrm{eff}<50\mu m}$ (μm) | 11 [10 / 13] | 13 [6 / 28] |
| $\widetilde{\mathrm{LWC}}_{<50\mu m}$ (g m$^{-3}$) | 0.06 [0.02 / 0.11] | (0.02 [0.01 / 0.12])×10$^{-3}$ |
| $\tilde{N}_{>50\mu m}$ (cm$^{-3}$) | (11.45 [4.83 / 21.03])×10$^{-3}$ | (2.44 [0.21 / 11.64])×10$^{-3}$ |
| $\tilde{D}_{\mathrm{eff}>50\mu m}$ (μm) | 1173 [879 / 1675] | 747 [473 / 1479] |
| $\widetilde{\mathrm{IWC}}_{>50\mu m}$ (g m$^{-3}$) | 0.09 [0.05 / 0.16] | (21.52 [3.74 / 108.98])×10$^{-3}$ |

### 3.1.2 Influence of environmental conditions

In the following, we relate the distinct cloud microphysical properties, cloud droplets versus haze droplets, to the prevailing environmental conditions that drive the formation of the respective cloud regimes. The meteorological parameters $T$, RH$_\mathrm{w}$,

and RH$_\mathrm{ice}$, measured in the cloud regimes classified as ice, MPH, MPC, liquid, and aerosol, are shown in Fig. 5. The data include all in-situ measurements from vertical flight profiles through the ABL that are temporally and spatially correlated with the meteorological data measured by the dropsondes (Sect. 2.3). Since mixed-phase clouds only exist at temperatures below 0 °C, all mixed-phase clouds are detected at ice supersaturated conditions. These results are in line with expectations, as classic stratiform mixed-phase clouds consist of ice particles and water droplets, with the ice crystals growing at the expense of the

liquid water droplets due to water vapor deposition. A persistent Wegener-Bergeron-Findeisen process in the mixed-phase clouds implies saturation with respect to water in order to compensate the water mass transfer from the liquid water droplets to the ice crystals. Consequently, a mixed-phase is unexpected to be measured at a humidity far below saturation with respect to water, because of the short relaxation time (only a few seconds) of liquid water droplets (Korolev et al., 2017). Previous studies, such as Korolev and Isaac (2006) and Costa et al. (2017), have confirmed that the water vapor in mixed-phase clouds is

close to saturation over water. Due to the greater environmental sensitivity of the liquid water phase compared to ice crystals,



pure liquid clouds display a similar distribution to that of MPCs in Fig. 5. However, liquid clouds are observed more frequently at colder temperatures, resulting in slightly higher supersaturation levels with respect to ice. For in-situ measurements of cloud regimes in absent of liquid water droplets, the ambient meteorological conditions are significantly different. The ice, aerosol, and MPH regimes are observed in notably drier conditions compared to the liquid and MPC regimes. These drier conditions

preclude measurements at $RH_w = 100\%$ in nearly all cases. However, ice and MPH regimes often display supersaturation with respect to ice, promoting the growth and persistence of ice crystals. A minor fraction of ice regimes exists in ice subsaturated conditions at $RH_{ice} < 100\%$, which is consistent with previous observations of ice crystals under non-equilibrium conditions (Korolev and Isaac, 2006; Voigt et al., 2017; De La Torre Castro et al., 2023; Dekoutsidis et al., 2023). Frequent observations of MPH and aerosol regimes under subsaturated conditions with respect to water support the hypothesis that the small particles

detected by the CDP are not cloud droplets. While ice crystals can persist in subsaturated air masses due to their extended phase relaxation time, comparable to the lifespan of an entire ice cloud (Krämer et al., 2009; Rollins et al., 2016; Korolev et al., 2017), small water droplets would evaporate immediately under such conditions. However, MPH is the most frequently observed cloud regime during the HALO-(AC)[3] campaign (see Sect.3.2) and during the AFLUX spring campaign in 2019 (Moser et al., 2023b), suggesting the presence of a stable thermodynamic condition. This thermodynamic stability of MPH in water

subsaturated conditions can be attributed to haze droplets composed of water and dissolved SSA. According to Köhler theory (Yau and Rogers, 1996; Laaksonen et al., 1998), these droplets achieve thermodynamic equilibrium with their surroundings at relative humidities below water saturation due to the Raoult effect. As a result, haze droplets can persist in subsaturated conditions, where pure water droplets would rapidly evaporate. Such haze droplets are often referred to as non-activated cloud droplets (Yau and Rogers, 1996; Hobbs and Wallace, 2006).

The size of a haze droplet is determined by the hygroscopic growth factor $g_e$. This factor, which represents the ratio between the radius of the haze droplet and the dry radius of the dissolved sea salt within the haze droplet, can be derived from the Köhler equation as given in Yau and Rogers (1996). Assuming the Kelvin term is negligible, the growth factor is expressed as:

$$g_e = \left( \frac{i_v M_V}{\rho_w M_S (1 - RH)} \cdot \rho_s \right)^{1/3} \tag{2}$$

with the assumptions that van't Hoff factor $i_v = 2$, molar mass of water ($H_2O$) $M_V = 18.02\,\mathrm{g\,mol^{-1}}$, density of water $\rho_w$

$= 1000\,\mathrm{kg\,m^{-3}}$, molar mass of the solute (NaCl) $M_S = 58.44\,\mathrm{g\,mol^{-1}}$, and density of the solute $\rho_s = 2160\,\mathrm{kg\,m^{-3}}$. With a measured median relative humidity within the MPH regime of $RH = 87.3\%$ (25th percentile = 84.8% / 75th percentile = 92.3%), the resulting hygroscopic growth factor for haze droplets in the MPH regime is 2.2 (2.1/2.6). Based on the calculated hygroscopic growth factor and the observed haze droplet sizes ranging from 3 μm to 6 μm, the estimated dry diameter of the solute is between 1 μm and 3 μm. Despite ambient conditions being subsaturated with respect to water, relative humidity

remained above the deliquescence point for NaCl ($RH_w = 74\%$; Zieger et al., 2017), such that sea salt particles are constantly in a dissolved state. Phase hysteresis effects including efflorescence and deliquescence are not relevant in this context (Tang et al., 1997; Zieger et al., 2017).



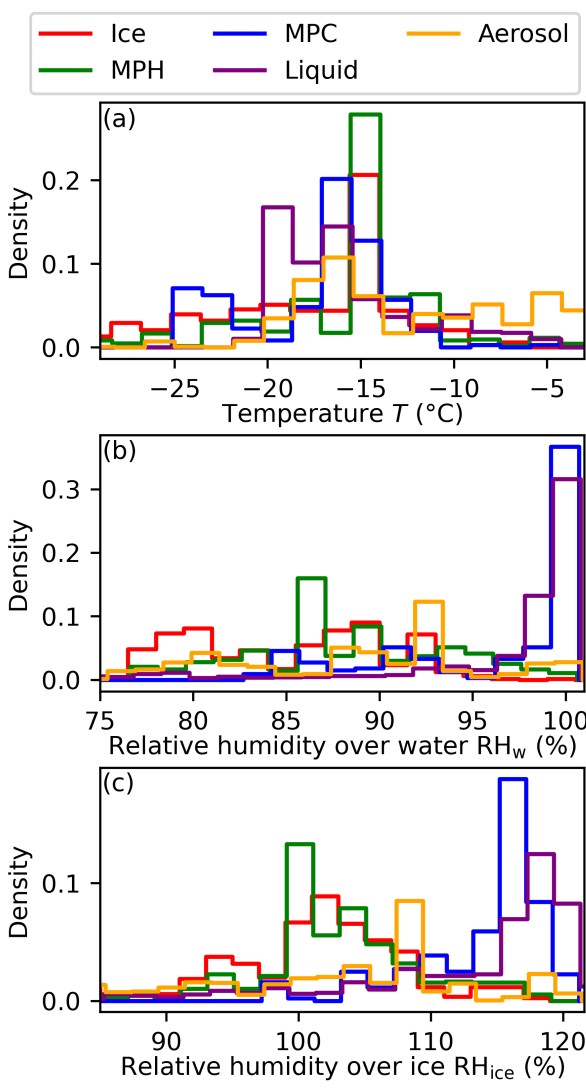

**Figure 5.** Normalized frequency distribution of the different cloud regimes as a function of environmental conditions such as temperature (a), relative humidity over water (b) and relative humidity over ice (c).





### 3.1.3   Chemical analysis and the origin of the haze particles

The microphysical description of mixed-phase haze assumes the presence of SSA in the Arctic ABL. It is well established
that sodium chloride (NaCl) from sea spray emission is one of the dominant locally-emitted aerosol types in the marine Arctic
from late winter through spring (Kirpes et al., 2018, 2019; Chen et al., 2022; Heutte et al., 2025). This assumption is further
supported by measurements from the ALABAMA instrument. Figure 6 presents the particle fraction derived from all available
ALABAMA measurements below 300 m in the ABL, where the thermodynamic cloud phase classification algorithm identified
either MPH or aerosol conditions. In addition, only sampling periods outside clouds were considered, as only a few cloud
particle residues were collected for these thin-cloud events. Figure 6 demonstrates that, in the size range above $\sim 700$ nm, the
majority of particles are SSA. It must be considered, that the particle size obtained by the ALABAMA is given in vacuum-
aerodynamic diameter ($d_{\mathrm{va}}$). Assuming a shape factor for cubic or agglomerates particles and a density of $\rho_{\mathrm{s}} = 2160$ kg m$^{-3}$
for the NaCl-containing particles (Zelenyuk et al., 2006), the volume-equivalent diameter ($d_{\mathrm{ve}}$) would be approximately 0.5
or 0.65 times the vacuum-aerodynamic diameter $d_{\mathrm{va}}$, respectively. This would result in particles sizes in the range between
$\sim 0.4$ μm and 2 μm in terms of $d_{\mathrm{ve}}$ for the observed SSA. Thus, there is an overlap of the dry diameters of the haze particles
from the hygroscopic growth assumption and the SSA observed by the ALABAMA. Given a relative humidity below 10% for
most of the time within the sampling lines, the SSA should be crystallized arriving at the ALABAMA inlet. However, only
few particles were analyzed by the ALABAMA above 3 μm, which might be due to a combination of the low transmission
efficiency of the CVI inlet line and the reduced detection efficiency of the ALABAMA in this size range. Nevertheless, the
domination of SSA at larger particle sizes and the potential of hygroscopic growth for dry NaCl particles to sizes around 5 μm,
suggest wet SSA contributing to the observed haze particles. Similar observations of large SSA have been reported in other
marine boundary layer studies, highlighting that such particles can act as giant cloud condensation nuclei even outside the
Arctic (Gonzalez et al., 2022). Ji et al. (2025) hypothesize that giant cloud condensation nuclei mixed within mixed-phase
clouds can modify the balance between ice sublimation and droplet growth, with potential implications for the cloud's lifetime.

The SSA are typically found within the coarse mode of the aerosol size distribution (Lohmann et al., 2016), which cannot
be fully resolved by the CDP due to its lower detection limit of 3 μm. Therefore, in the following analysis, OPC measurements
are utilized to investigate the haze droplets with a particular focus on the influence of the underlying surface conditions on
their properties. Previous studies have shown that over the ocean, SSA are primarily generated by wave-breaking processes
(Blanchard, 1989), whereas over sea ice, mechanisms such as blowing snow or frost flowers contribute to their emission (Yang
et al., 2008; Seguin et al., 2014; Xu et al., 2016; Huang and Jaeglé, 2017; Willis et al., 2018; Marelle et al., 2021; Lapere et al.,
2024). Figure 7 displays the boxplots of particle number concentration $N$ (a) and effective diameter $D_{\mathrm{eff}}$ (b) derived from OPC
measurements over the sea ice and the open ocean. Only particles with diameters exceeding 500 nm within the ABL (below
300 m) and within aerosol or mixed-phase haze conditions are considered. For each day with sufficient measurements over the
sea ice and the ocean ($> 100$ s respectively), the data from these two surface types are displayed. The boxes labeled as "total"
include the data from all campaign days. A significance test is applied to assess whether the measurements over sea ice and
the ocean differ significantly on a given day. The Mann-Whitney U test is performed, using a significance level of $p < 0.05$.



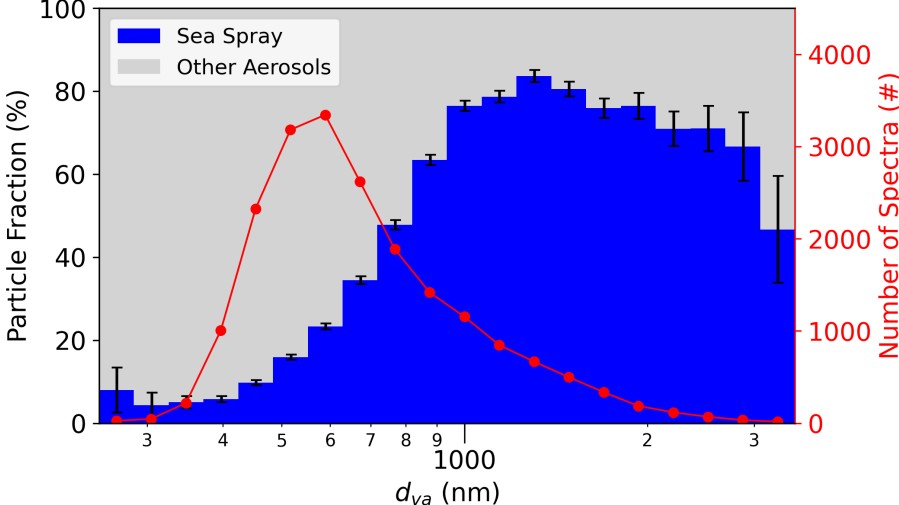

**Figure 6.** Number fraction of sea spray particles (blue) containing NaCl ion signals and other aerosol particles (gray) as a function of the particle size given by the vacuum-aerodynamic diameter ($d_{va}$). The number fraction for each logarithmic size bin was calculated with respect to the total number of spectra collected within the respective bin (red markers). The given uncertainties result from binomial statistics and are further described in Appendix B.

The data sets that have been compared are connected with a dotted line in Fig. 7. On all days except 1 April, the surface type has a significant effect on $N$. The effective diameter $D_{\mathrm{eff}}$ differs significantly between the two surfaces on six days, while no significant difference is observed on 26 March, 29 March and 9 April. In the case of $N$, six days show a significantly higher
particle number concentration over sea ice compared to the ocean, indicating with high confidence that $N$ is enhanced over sea ice. Only two exceptions are found: On 5 April and 10 April, $N$ is higher over the open ocean than over sea ice. Thus, the overall OPC dataset shown as the "total" boxplot robustly represents the observed increase in $N$ over sea ice relative to the open ocean. The measurements of $D_{\mathrm{eff}}$ show that on 20 March, $D_{\mathrm{eff}}$ is significantly higher over sea ice, whereas on five days, the $D_{\mathrm{eff}}$ is significantly larger over the open ocean compared to sea ice. Consequently, the total boxplot of $D_{\mathrm{eff}}$ represents the
increase over the ocean relative to the sea ice.

Although the optical properties of MPH measured with the Polar Nephelometer indicate the presence of ice and liquid water (Moser et al., 2023b), it cannot be completely excluded that some haze droplets in MPH could be small, spherical ice crystals. However, previous in-situ aerosol measurements in the Arctic have shown that the number concentration of ice nucleating particles (INP) is mostly well below $10^4\,\mathrm{m}^{-3}$ at temperatures higher than -20°C (Dietel et al., 2024). These measured INP
number concentrations are significantly smaller (at least a factor 10, more likely a factor $10^2$ - $10^5$ compared to Dietel et al. (2024)) than the measured number concentrations of haze droplets in MPH. Therefore, the contribution of potentially glaciated haze droplets in MPH is negligible.

The Arctic low-level mixed-phase haze presented here is not be confused with the so-called Arctic haze. Arctic haze is a phenomenon observed predominantly in winter, characterized by high aerosol mass concentrations in the Arctic (Rahn et al.,

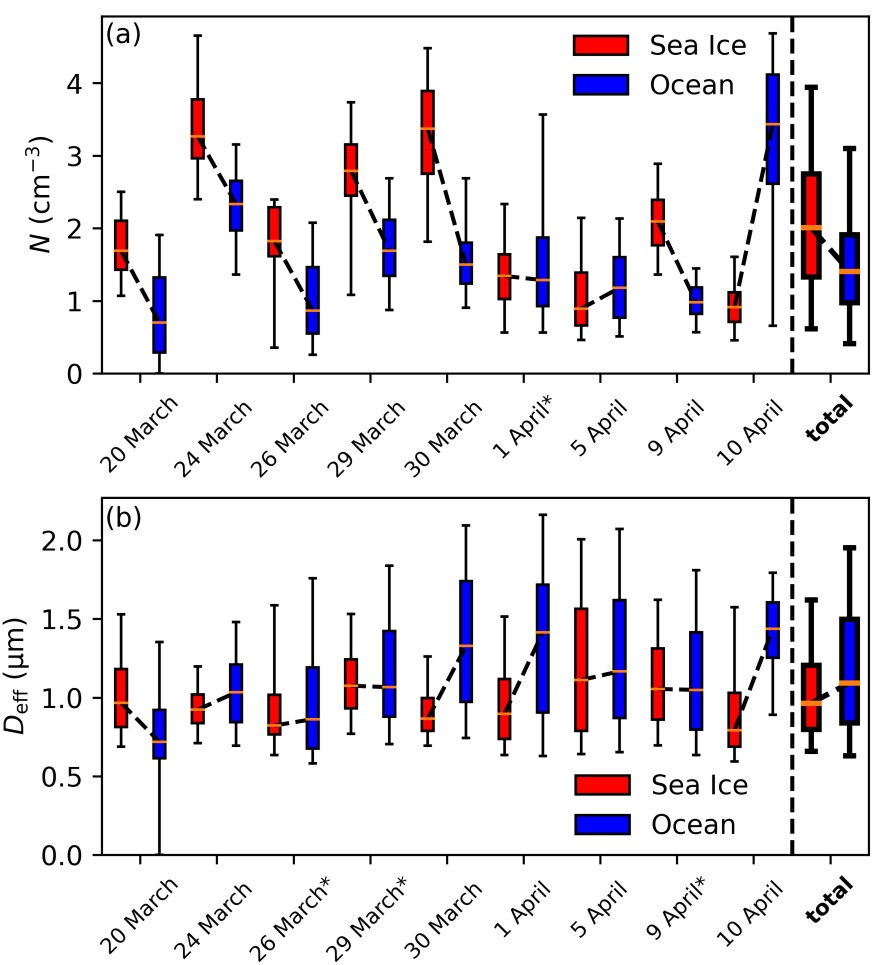

**Figure 7.** Overview of the $N$ (a) and the $D_{\text{eff}}$ (b) derived from the OPC. The boxplots include the median, upper, and lower quartiles, and the whiskers give the 5th and 95th percentile. Only particles with diameters larger 500 nm inside the ABL and during aerosol or mixed-phase haze conditions are considered. Statistically insignificant values are marked with an asterisk.




1977; Radke et al., 1984; Willis et al., 2018; Schmale et al., 2021). It mainly consists of aerosols from anthropogenic emissions
transported over long distances from mid-latitudes to the Arctic, and is found at all altitudes in the troposphere (Quinn et al.,
2007). In contrast, the haze particles investigated in this study primarily consist of SSA from local marine sources and are
associated with low altitudes.

### 3.2   Vertical thermodynamic phase distribution of Arctic low-level clouds during HALO-(AC)[3]

For the analysis of the vertical distribution of thermodynamic cloud phases, all vertical flight legs with sufficient in-situ data
are considered. Figure 8 presents the frequency distribution of detected thermodynamic phases in low-level cloud regimes
($< 1000$ m). With a frequency of 34 %, the MPH is the most common observed regime, followed by the aerosol regime with
27 % and the liquid water phase with 23 %. The ice and MPC regimes are significantly less frequent, accounting for 12 % and
4 %, respectively. This distribution of thermodynamic phases is consistent with observations of Arctic low-level mixed-phase
clouds during the AFLUX campaign in spring 2019 (see Fig. 9 in Moser et al., 2023b). The major differences are that fewer pure
aerosol regimes were detected during AFLUX, and the pure liquid water phase was observed slightly less frequently. Despite
differences in measurement strategies between AFLUX and the dataset presented in Fig. 8, as well as variations in the years of
observation, the distributions align well. This consistency further supports the representativeness of this thermodynamic phase
distribution for Arctic spring conditions in low-level altitude.

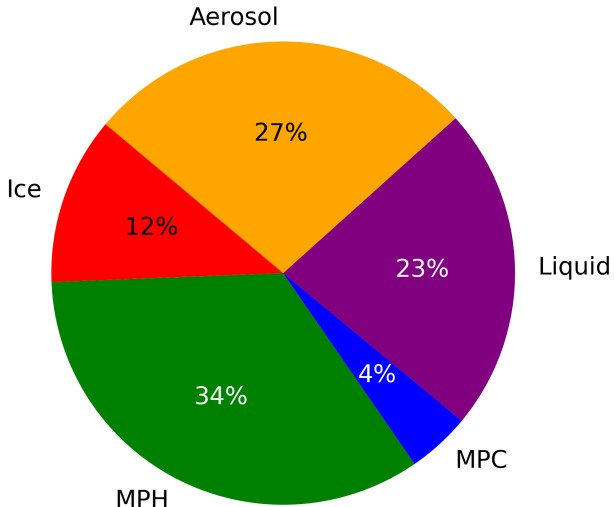

**Figure 8.** The fraction of different cloud regimes detected during all vertical flight legs below 1000 m during the HALO-(AC)[3] campaign.

Figure 9 illustrates the vertical distribution of thermodynamic cloud phases as a function of normalized altitude (Eq. 1) based
on the height of the ABL for three different surface conditions. (a,b) includes all in-situ data collected from vertical flight legs
over sea ice, (c,d) shows in-situ data from vertical flight legs over the MIZ, and (e,f) represents in-situ data from vertical flight
legs over the ocean. Across all surface types, similar patterns in the vertical distributions can be observed. At the top of the ABL,





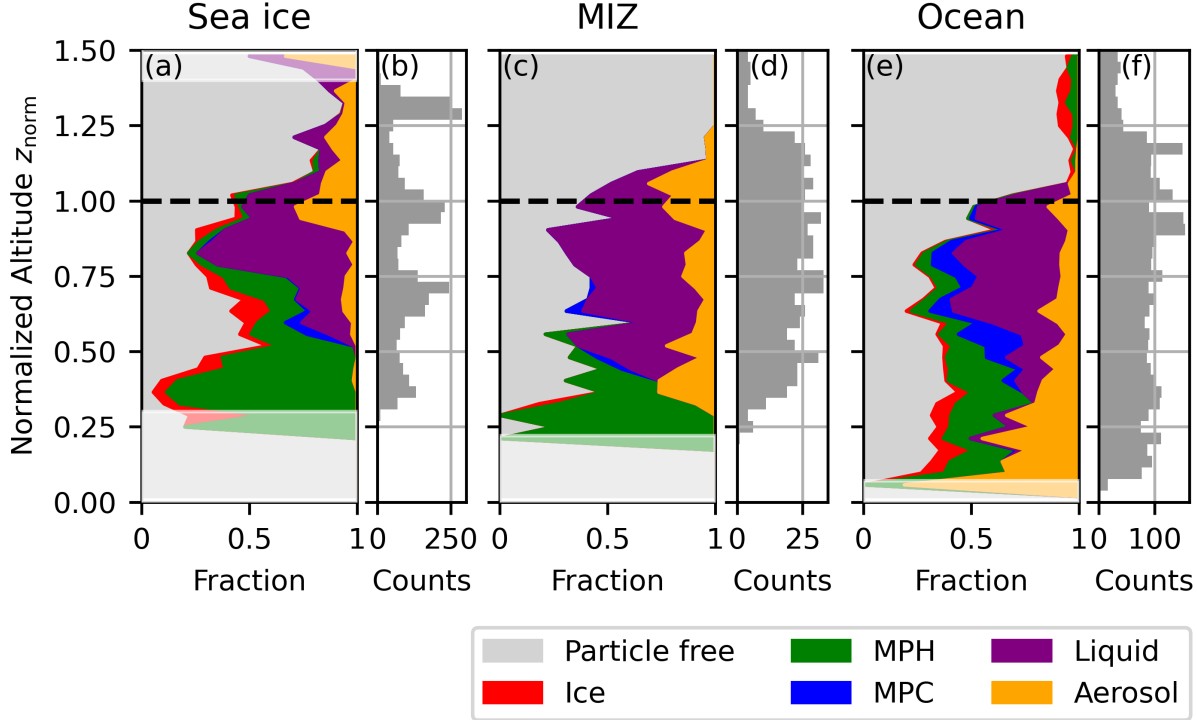

**Figure 9.** Distribution of the thermodynamic cloud phases derived from the in-situ cloud probe data, shown in normalized altitude. (a) shows all vertical profiles over sea ice, (c) in the marginal sea ice zone, and (e) over the open ocean. (b), (d), and (f) show the number of measurement points at the respective altitude.

clouds are most frequently in a liquid state, with pure aerosol measurements occurring less frequently than the liquid regime at
this altitude. Below the ABL top, the frequency of the liquid regime continues to increase in all cases, while simultaneously, the occurrence of MPH rises with decreasing altitude. MPH becomes the dominant thermodynamic phase at lower altitudes over sea ice and in the MIZ. The regime MPC is primarily observed in the transition region, where the dominant phase shifts from liquid to MPH with decreasing altitude. Above the ABL top, cloud and aerosol measurements become significantly less frequent. Some specific deviations from these general patterns are apparent. While no pure ice phase is detected in the MIZ, it
is consistently present throughout the entire ABL over sea ice and is observed only at lower altitudes (<0.5 normalized altitude) over the ocean. Unlike over sea ice and the MIZ, the frequency of aerosol measurements increases with decreasing altitude over the ocean, making aerosols the most frequently detected regime at the lowest measurement levels over the ocean. Furthermore, it is observed that the frequency of MPCs decreases from the ocean towards the sea ice. The lower occurrence of MPCs over sea ice compared to the open ocean has also been reported by Mioche et al. (2015, 2017). Possible explanations include the reduced
moisture availability over sea ice, resulting from limited updraft-driven supersaturation necessary to sustain supercooled liquid water droplets, as well as enhanced glaciation efficiency in colder and more stable atmospheric conditions. In particular,





weak updrafts over sea ice often fail to maintain supersaturation with respect to water, causing rapid droplet evaporation under supersaturated conditions with respect to ice, whereas stronger updrafts over open ocean favor the persistence of MPCs (Korolev and Field, 2008; Costa et al., 2017). Also ice nucleating particles are expected to be less pronounced over the sea ice
(Dietel et al., 2024).

Compared to the 33 individual profiles analyzed in this study, the thermodynamic phase distribution shown in Figure 9 represents the overall characteristics well. In almost all individual cases, a stratification with a liquid layer at the cloud top, an MPC layer beneath it, and an MPH regime in the lower part of the cloud can be observed, consistent with the vertical thermodynamic phase pattern documented in marine cold-air outbreaks by Schirmacher et al. (2024). However, it is important
to note that in six individual cases, the MPH regime was detected without any liquid (liquid regime or MPC regime) layer present in the cloud profile. This suggests that the MPH regime can persist even in the absence of a liquid and MPC layer, likely resulting from the evaporation of the liquid water phase, leaving only the MPH layer behind.

As an example to illustrate the visual appearance of the MPH regime, Figure 10 (a) presents a photograph taken from the cockpit just above the cloud layer, and (b) showing the lower part of a low-level cloud layer. The vertical analysis during this
ascent indicates that at the time the photograph in (b) was taken, the prevailing regime was MPH. Further up in the cloud, a layer of MPC was observed, followed by a pure liquid layer at the cloud top, which is visible from above in (a). The visibility within an MPH, as seen in (b), is subjectively high, reaching up to several hundred meters. A slight haze is observed, but no whiteout conditions occur, as seen in MPC and liquid layers. Consistent with the microphysical properties, the optical thickness of MPH is significantly lower than that of MPC.

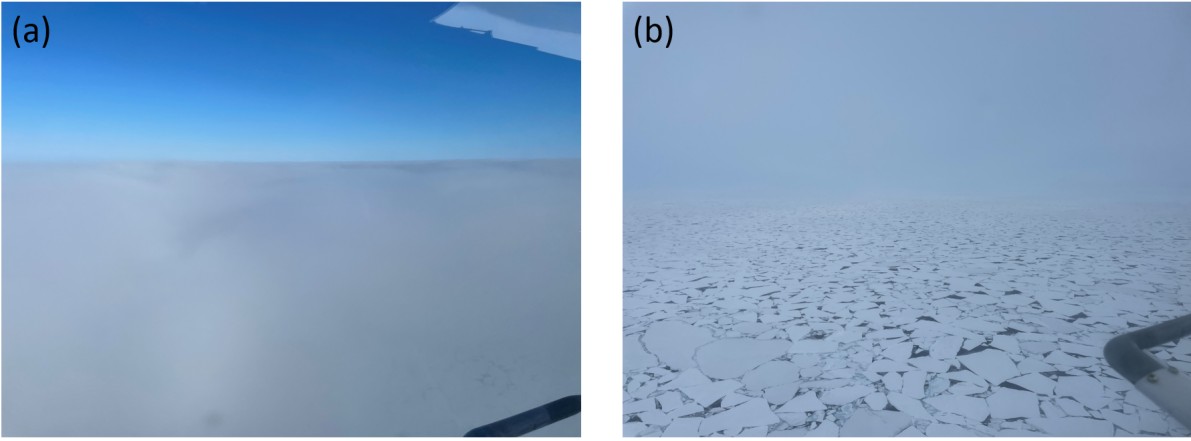

**Figure 10.** Pictures taken during an ascent through an Arctic low-level cloud on 10 April 2022, which consisted of an MPH regime at the bottom, an MPC regime in the middle, and a liquid layer at the cloud top. (b) was taken at 11:53 UTC within the MPH regime, while (a) was taken at 11:55 UTC just above the liquid layer.





## 4 Summary and conclusion

During the aircraft field campaign HALO-(AC)[3], we collected a comprehensive in-situ dataset of microphysical cloud properties below 1 km altitude above sea ice, MIZ and open ocean in the Fram Strait. A particular focus was on clouds exhibiting mixed-phase conditions. Primary results are listed below and schematically summarized in Fig. 11:

- Thermodynamic phase analyses of clouds in a mixed-phase state indicate the presence of both liquid water and ice phases. However, based on their microphysical properties, we distinguish between a classic mixed-phase cloud (MPC) and a mixed-phase haze (MPH) regime.

- While an MPC consists of a mixture of supercooled liquid water droplets and larger ice particles, the MPH is composed of unactivated haze droplets, representing solutions of water and saline aerosols, mixed with ice crystals.

- The median $N$ and median CWC in MPCs are significantly higher compared to those in MPH, with median number concentrations increased by a factor of 1000 and cloud water content by a factor of 8. This difference is also reflected in the substantially reduced optical thickness ($\tau$) observed for MPH.

- In MPCs, the coexistence of liquid water droplets and ice crystals requires sustained saturation with respect to water and supersaturation with respect to ice. In contrast, MPH exist in regions with $RH_w < 100\,\%$, where the persistence of haze droplets is described by unactivated cloud droplets according to the Köhler theory.

- Chemical mass spectrometric analyses identify a dominant portion of sea salt aerosols (SSA) in the size range between $0.4\,\mu m$ and $2\,\mu m$. This size range overlaps with the estimated dry diameter of the haze droplets with the assumption of NaCl composition suggesting that the observed haze droplets consist of SSA.

- Measurements with the OPC reveal a significantly increased number concentration of haze particles over the sea ice compared to the open ocean. This suggests that aerosol-generating processes over sea ice, such as blowing snow and frost flowers, contribute more to aerosol production than wave-breaking mechanisms over the ocean.

- The analysis of the vertical thermodynamic phase distribution in Arctic boundary layer clouds reveals that the most common structure follows a pattern where liquid clouds are found at the top of the ABL, followed by a classic mixed-phase cloud, and an MPH layer beneath. In some cases, pure MPH regimes without a liquid water phase are also observed within the ABL.

The findings emphasize the need of distinguishing between different types of mixed-phase regimes within Arctic boundary layer clouds. The frequency and persistence of MPH regimes imply the importance of representing this cloud regime in climate and weather models. This is essential for assessing the role of low-level clouds in the Arctic radiation budget and quantifying their feedback mechanisms in the region most affected by anthropogenic climate change.



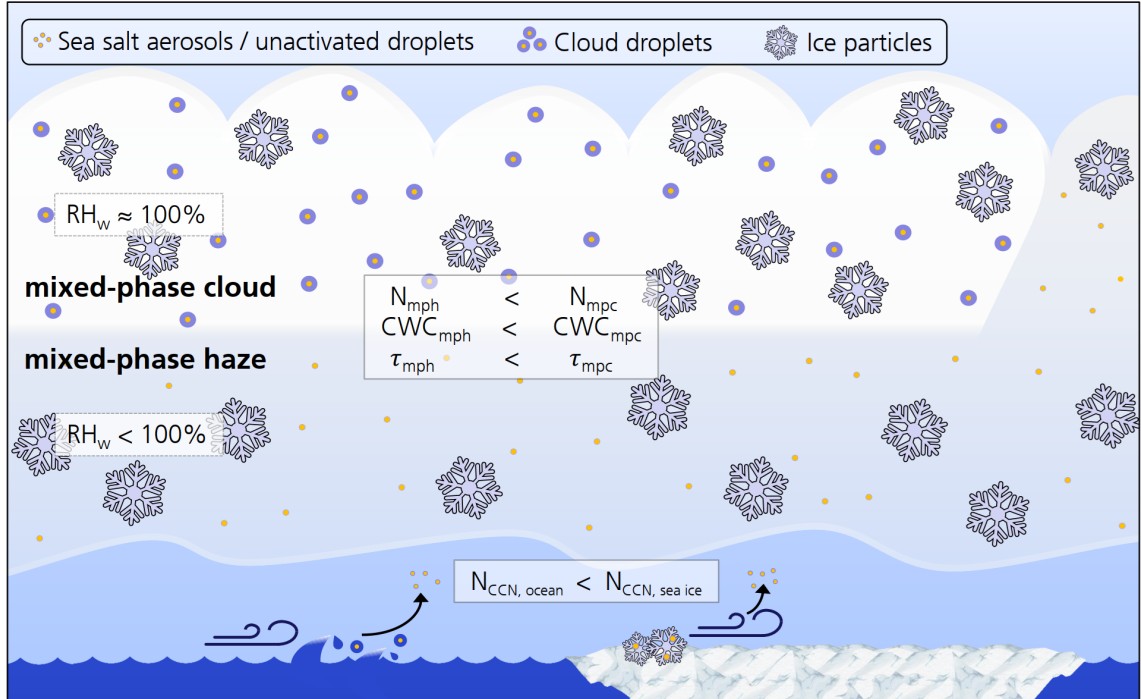

**Figure 11.** Summary: Schematic representation of the microphysical properties of mixed-phase haze and mixed-phase clouds.

*Data availability.* Processed in-situ cloud, in-situ aerosol, and dropsonde data from the HALO-(AC)[3] campaign are freely available via the world data center PANGAEA (Eppers et al., 2023a, b; George et al., 2024; Moser et al., 2023a; Mertes and Wetzel, 2023). Data can be easily reproduced and analyzed by the python package *ac3airborne* (Mech et al., 2022b) including a package for flight segmentation (Risse et al., 2022), where each research flight is split up into logical parts like ascends, descends, specific patterns for in-situ probing, etc.. Raw in-situ cloud data recorded by the CDP, CIP and PIP are archived at the German Aerospace Center and are available on request. Raw data by the OPC and the ALABAMA are available from O. Eppers on request. Figures have been design with the python software Pylustrator (Gerum, 2020).

*Author contributions.* MMo conducted the analysis and wrote the manuscript. CV supervised the study and provided intensive feedback on the manuscript. MMo, JL, EDLTC, JM, RD, GM and OJ have been responsible for the in-situ cloud probes and performed the measurements. OE, HCC, PJ, JS, BW and SM performed the aerosol measurements during the campaign. OE and HC analyzed the ALABAMA and OPC data. MMo, CV, JS, SB, MK, MMe, CL, SC, AE, AH, and MW conceived the flight experiments. All authors reviewed the manuscript and added valuable suggestions to the final draft.



*Acknowledgements.* MMo, JM, CV, OE, HCC, PJ, JS and EDLTC gratefully acknowledge funding by the Deutsche Forschungsgemeinschaft (DFG, German Research Foundation) through TRR 301 "TPChange" (Project ID 428312742) and SPP 1294 "HALO" (Project IDs 522359172 and 442647984 and 316646266). MW, MK, SM, BW, MMe, SC, CL, AE and AH gratefully acknowledge funding by the DFG through TRR 172 "ArctiC Amplification: Climate Relevant Atmospheric and SurfaCe Processes, and Feedback Mechanisms (AC)[3]" (Project ID 268020496).

OJ, RD and GM gratefully acknowledge support from the Centre National d'Études Spatiales (CNES) via the project "Expecting Earth-Care, Learning from A-Train" (E-ECLAT) and from CNRS/INSU through the INSU-LEFE program. Their research activities are a contribution to the (MPC)[2] project, supported by the Agence Nationale de la Recherche under the Grant ANR-22-CE01-0009. JL gratefully acknowledges support from the European Union's Horizon 2020 research and innovation programme (SENS4ICE, grant agreement no. 824253). SB acknowledges funding from internal sources of the Max-Planck-Society.

We gratefully acknowledge the AWI flight crew and technicians for their outstanding support during HALO-(AC)[3].

*Financial support.* This reseach has been supported by the Deutsche Forschungsgemeinschaft (TRR 301, grant no. 428312742; SPP 1294, grant nos 522359172 and 316646266 and 442647984; TRR 172, grant no. 268020496). The Centre National Etudes Spatiales ("Expecting Earth-Care, Learning from A-Train" (E-ECLAT) and CNRS via its program INSU-LEFE). The Agence Nationale de la Recherche (grant no. ANR-22-CE01-0009). The European Union's Horizon 2020 research and innovation programme (grant no. 824253).

*Competing interests.* The contact author has declared that none of the authors has any competing interests.





## Appendix A: Metadata for vertical profiles

**Table A1.** Table of flight legs, including the start and end time of each vertical flight leg, the corresponding dropsonde ID for the collocated dropsonde, the mean sea ice concentration below the flight leg, the origin of the air mass prior to measurement, the altitude of the ABL, and the temperature at the ABL height.

| Start Time | End Time | Dropsonde ID | SIC (%) | Air Mass Origin | ABL Height (m) | $T$ at ABL top (°C) |
|---|---|---|---|---|---|---|
| 20 March 2022, 11:56:38 | 12:09:03 | P5_RF01_01 | 0.0 | sea ice | 864 | -17.15 |
| 20 March 2022, 12:10:15 | 12:13:00 | P5_RF01_02 | 0.0 | sea ice | 665 | -17.31 |
| 20 March 2022, 12:13:50 | 12:16:35 | P5_RF01_08 | 22.77 | sea ice | 599 | -17.14 |
| 20 March 2022, 12:18:42 | 12:21:22 | P5_RF01_03 | 86.83 | sea ice | 535 | -18.31 |
| 20 March 2022, 12:21:55 | 12:23:46 | P5_RF01_03 | 97.54 | sea ice | 505 | -19.51 |
| 20 March 2022, 12:24:01 | 12:26:14 | P5_RF01_03 | 99.85 | sea ice | 479 | -20.02 |
| 20 March 2022, 12:26:24 | 12:28:12 | P5_RF01_07 | 98.91 | sea ice | 483 | -20.11 |
| 20 March 2022, 12:28:21 | 12:29:56 | P5_RF01_07 | 98.58 | sea ice | 398 | -19.17 |
| 20 March 2022, 12:53:57 | 12:57:12 | P5_RF01_05 | 75.94 | sea ice | 295 | -18.27 |
| 20 March 2022, 13:44:18 | 13:46:13 | P5_RF01_04 | 100.0 | sea ice | 379 | -18.65 |
| 20 March 2022, 13:48:52 | 13:50:47 | P5_RF01_07 | 99.18 | sea ice | 528 | -20.81 |
| 20 March 2022, 13:51:02 | 13:53:50 | P5_RF01_07 | 99.76 | sea ice | 550 | -20.24 |
| 20 March 2022, 13:59:57 | 14:03:01 | P5_RF01_08 | 32.89 | sea ice | 631 | -17.37 |
| 20 March 2022, 14:03:16 | 14:07:44 | P5_RF01_08 | 0.0 | sea ice | 850 | -17.26 |
| 20 March 2022, 14:08:10 | 14:12:15 | P5_RF01_02 | 0.0 | sea ice | 857 | -16.91 |
| 20 March 2022, 14:41:58 | 14:45:42 | P5_RF01_09 | 0.0 | sea ice | 761 | -16.86 |
| 20 March 2022, 15:20:15 | 15:34:12 | P5_RF01_12 | 3.14 | sea ice | 853 | -15.83 |
| 22 March 2022, 13:54:53 | 13:56:11 | P5_RF03_04 | 93.29 | sea ice | 186 | -26.19 |
| 22 March 2022, 14:19:36 | 14:30:27 | P5_RF03_07 | 14.45 | sea ice | 349 | -22.98 |
| 29 March 2022, 14:59:48 | 15:10:59 | P5_RF07_05 | 0.0 | sea ice | 511 | -13.70 |
| 30 March 2022, 11:18:10 | 11:30:32 | HALO_RF11_09 | 65.07 | sea ice | 183 | -17.70 |
| 30 March 2022, 13:23:22 | 13:24:49 | HALO_RF11_20 | 0.0 | sea ice | 569 | -12.44 |
| 30 March 2022, 13:24:58 | 13:28:59 | HALO_RF11_20 | 0.0 | sea ice | 581 | -12.45 |
| 5 April 2022, 10:40:14 | 10:46:21 | P5_RF11_02 | 0.0 | ocean/land | 694 | -9.46 |
| 10 April 2022, 10:53:13 | 11:35:23 | P5_RF13_14 | 97.08 | sea ice | 181 | -15.72 |
| 10 April 2022, 11:42:34 | 11:44:56 | P5_RF13_05 | 96.26 | ocean/land | 177 | -16.34 |
| 10 April 2022, 11:45:05 | 11:46:44 | P5_RF13_14 | 98.57 | ocean/land | 173 | -15.89 |
| 10 April 2022, 11:48:49 | 11:50:32 | P5_RF13_04 | 93.69 | ocean/land | 292 | -13.24 |
| 10 April 2022, 11:50:47 | 11:53:01 | P5_RF13_04 | 95.53 | ocean/land | 317 | -11.66 |
| 10 April 2022, 11:53:11 | 11:55:18 | P5_RF13_04 | 80.95 | ocean/land | 373 | -7.54 |
| 10 April 2022, 11:55:33 | 11:57:47 | P5_RF13_15 | 38.16 | ocean/land | 348 | -10.38 |
| 10 April 2022, 12:06:00 | 12:11:10 | P5_RF13_12 | 0.0 | ocean/land | 1018 | -9.37 |
| 10 April 2022, 12:14:20 | 12:18:13 | P5_RF13_11 | 0.0 | ocean/land | 937 | -9.00 |





## Appendix B: Uncertainty analysis for the particle fraction of NaCl

The absolute uncertainty of the ALABAMA particle fraction for each bin ($\sigma_{\mathrm{PF}}^{\mathrm{abs}}$) is calculated using binomial statistics (e.g., Köllner et al., 2017; Köllner et al., 2021):

$$\sigma_{\mathrm{PF}}^{\mathrm{abs}} = \frac{\sqrt{N_{\mathrm{hits}} \cdot \mathrm{PF} \cdot (1 - \mathrm{PF})}}{N_{\mathrm{hits}}}, \tag{B1}$$

with the number of successfully ionized particles ($N_{\mathrm{hits}}$) by the desorption laser of the ALABAMA and the particle fraction of the respective particle type (PF).



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
