# Peer review of "The Arctic Low-Level Mixed-Phase Haze Regime and its Microphysical Differences to Mixed-Phase Clouds"

_EGUsphere, 2025_

## Referee Comment (RC1)

**Review for "The Arctic low-level mixed-phase haze regime and its microphysical differences to mixed-phase clouds" by Moser et al., ACPD**

**Summary**

The study presents arguments for distinguishing between Arctic mixed-phase clouds and Arctic mixed-phase haze. The latter occurs at water subsaturated conditions and consists of non-activated haze droplets and ice crystals. Clear cloud microphysical differences are identified and support the presence of the Arctic mixed-phase haze. The authors combine several measurements from the HALO-(AC)[3] campaign, which also allows them to characterize the vertical structure of these clouds in the Arctic boundary layer as well as to assess the chemical composition of the measured aerosol particles. The authors did a nice job on combining all these data and draw new insights on Arctic mixed-phase haze. I recommend this study for publication after addressing my comments below.

**Major comments**

These are comments regarding the comprehension of the manuscript and include questions to help the clarification of the findings. The authors should consider addressing them.

- Line 45: The authors hint at the potential of MPH for the atmospheric radiation budget. However, throughout the manuscript no evidence is shown for that, and in the end, the authors state that MPH are optically thin (Line 394). I am wondering what is their importance? This also connects to how often this regime was identified, and especially with no liquid layer at cloud top. Because for MPC the liquid layer is the driver for the cloud top cooling and has the biggest impact of the radiative budget. Can you please elaborate on that more?

**Minor comments**

These are editorial comments helping to improve the formatting and readability of the manuscript. The authors should consider adding them.

- Line 13: While reading I was wondering if the MPH is a state instead of MPCs, but it appears that it is most often found beneath the MPC, which is stated here. However, there are cases of MPH without a MPC identified. I think the abstract would benefit from a clear statement on the occurrence and persistence of MPH.

- Line 21: "Many factors influencing Arctic amplification are discussed (Wendisch et al., 2023a)", is an incomplete sentence. Either you say "discussed in" or you rephrase it in such a way that it is clear that a list of possible reasons is following.

- Line 28: That MPCs are in a quasi-steady state is true, however, the reason is not only dynamical processes but rather a superposition of dynamics, turbulence, radiation, and cloud microphysics. This should be clearly stated, as it is still a challenge to fully understand how these clouds remain persistent given their metastable thermodynamic state.

- Line 36: replace "including those" by (e.g., ....).

- Line 44: using a threshold of LWC = 0.01 g m$^{-3}$

- Line 48: mixed-phase temperature regime needs to be defined (in numbers)

- Line 49: the Wegener-Bergeron-Findeisen process should be at least explain in 1-2 sentences, because it appears throughout the manuscript.

- Line 50: One-sentence paragraphs should be avoided.

- Introduction: While the abbreviations MPC (also it should be MPCs if it is used in the plural form) and MPH are introduced in the abstract, they are only defined later in the manuscript text, even though the regimes are discussed already in the introduction. Please double-check that you introduce the abbreviations at the earliest appearance and then use them throughout the text. This goes for several abbreviations, such as LWC and IWC as well.

- Line 66: what does "partially close proximity" mean? Can you just quantify it as you do it later in the text?

- Line 72: "had" to has

- Line 74: reference to Figure 1 when you introduce the location

- Line 84: CWC needs to be introduced, especially since I assumed you meant LWC. It should be clear that here the total water content (TWC) is meant. Also a short sentence on the threshold should be done, and not only referring to a previous study.

- Line 93: "N" needs to be defined

- Line 121: Why are legs from horizontal flights less significant?

- Line 126: Suggestion for reformulation: Moser et al. (2023b) introduced a method for determining ...

- Line 131: The sentence regarding the thresholds is somewhat confusing, and should be rephrased, such that word repetition is avoided.

- Table 1: the abbreviation were not introduced.

- Line 142: given that CAO and WAI are only used twice, consider not using the abbreviations and just write it out.

- Line 152: The classification ocean/land I find confusing as it should indicate that the air mass was mostly influenced by the ocean. In Line 182 the phrasing is also slightly different for the ocean/land mask, which makes it more confusing.

- Line 165: which data were now used?

- Line 168: the information in the brackets for SIC can be omitted as it was defined before

- Line 170: Suggestion for reformulation: ..., at which the minimum temperature occurs (i.e., minimum of temperature inversion).

- Figure 2: For all figures the colors should be revised such that they are color-vision-deficiency friendly (especially avoiding red and green lines). I was wondering, if the grey dropsonde profiles can be stratified by the origin as it was done for the mean value. You could use just a lighter color of the mean value to indicate that. Moreover, asterisks could be used to mark the mean ABL height.

- Line 172: First the figure should be introduced and then the interpretation should be done. Move the first sentence to after the sentence ending on Fig. 3.

- Line 177: The sentence "Consequently, ..." is a repetition of what was said in the sentence before. Can this be consolidated and maybe combined?

- Line 184: I do not see how Fig. 3 informs us on the vertical structure of the ABL? Do you mean the vertical extent? You only investigate the structure later in the manuscript.

- Line 187: Is it a surprise that the large-scale conditions define the temperature, but local processes determine the boundary layer dynamics? If so, this should be emphasized. If not, I would suggest to add references to contextualize your statement.

- Line 189: The definition of the normalized altitude should be introduced when it is actually used in Section 3.2.

- Line 203: "as" is missing and the explanation for combining the regimes for MPH could be supported by an appendix figure, if the authors want to.

- Line 220: The introduced regimes and abbreviations should be used explicitly and not converted back to full text (whole paragraph).

- Figure 4: add in the caption that the percentiles were calculated based on bootstrapping.

- Line 244: Why the reference to Moser et al., 2023b: It is just the sum of IWC and LWC, no? Here the CWC defintion needs to come earlier in the manuscript.

- Line 255: The Wegener-Bergeron-Findeisen process describes in principle the evaporation of cloud droplets and the growth of ice crystals. However, for cloud droplets to evaporate, water subsaturated conditions must exist, otherwise they are not evaporating. Then both cloud droplets and ice crystals would grow. So, the WBF process by itself does not imply water saturation, but rather the available cloud droplets. So one needs to be careful when arguing with the WBF process. Also I would argue that the WBF process is not persistent.

- Line 257: the word cloud is missing, but anyway the abbreviation MPC should be used. I agree that there is water saturation, but the WBF process is definitely not the only process helping to sustain that. This is too-short handed argued.

- Line 260: What do you mean by greater sensitivity of the liquid phase to the enviroment?

- Line 262: higher supersaturation than what?

- Line 265: Isn't it the defitnio of MPH and ice that RH < 100 %?

- Line 274: Are these really stable thermodynamic conditions? Wouldn't the existence of ice crystals with the haze droplets mean some metastable state?

- Line 314: Can you explain the process how GCCN are impacting ice sublmation and cloud droplet growth?

- Line 323: Why did you choose 300 m?

- Line 327: dotted should be dashed

- Line 336: I thought the MPH have water subsaturated conditions, how can there be liquid water present?

- Figure 7: The colors could be in agreement with Figure 3 and instead of using the same linestyle for separation the total, use maybe dotted.

- Figure 8: Can you turn the pie chart in such a way that "0 %" starts at the top of the circle? This way it is more intuititve.

- Figure 9: What is the white shading in a, c, and e?

- Line 376: Why is the glaciation efficiency higher in more stable conditions?

- Line 385: Here an assessment of the lifetime of MPH would be great.

- Line 399: Does not the mixed-phase state imply that we have liquid and ice crystals? So this is circular?

- Conclusion point 2 and 4: These are the same to me and could be combined.

- Line 406: The optical thickness was only assessed visually, no? Can you quantify it?

- Line 421: Are MPH really persistent? You did not show anything regarding lifetime?

---

## Community Comment (CC1)

Comment on:

**The Arctic Low-Level Mixed-Phase Haze Regime and its Microphysical Differences to Mixed-Phase Clouds**

**by Moser et al. (2025)**

The manuscript of Moser et al. (2025) (Moser-25, hereafter) is a very nice and interesting study that presents new insights into the formation of different particle size distributions (PSDs) of Arctic low-level mixed-phase clouds.

This commentary compares the study with that of Costa et al. (2017) (Costa-17, hereafter), who investigated PSD types occurring in mixed-phase clouds from five airborne campaigns in Arctic, mid-latitude and tropical regions within the temperature range 235 to 275 K. Costa-17's study is mentioned by Moser-25, but the results were not compared in detail - which I think is worth doing.

**Comparison of mixed-phase cloud PSDs in Moser-25 and Costa-17**

The measurements of Costa-17 were performed with almost identical cloud instrumentation and are therefore well comparable (see caption of Figure 1).

Moser-25 identified two types of mixed-phase clouds (see upper panel of Figure 1) in the temperature range 255-265 K, which they call MPC (mixed-phase clouds, blue) and MPH (mixed-phase haze, green). Costa-17 also found two mixed-phase cloud types (middle panels of Figure 1), named as 'Coexistence' (blue) and 'Large ice/WBF' (green) mixed-phase clouds. They show the PSDs in 5K temperature intervals, the range in which the Moser-25 measurements are located is circled in the legend (purple-reddish colors).

Comparison of the Moser-25 and Costa-17 PSDs in the overlapping temperature range shows the same features in both mixed-phase cloud types. Moreover, the MPC | 'Coexistence' (Moser-25 | Costa-17) and the MPH | 'Large ice/WBF' (Moser-25 | Costa-17) PSDs are very similar, which can be seen when comparing the blue and green horizontal lines, drawn to guide the eye. The reason why the largest ice crystals of Costa-17 are smaller than that of Moser-25 is that Costa-17 used 'area equivalent diameter' as size measure and Moser-25 used 'maximum dimension', and also that the sizerange of Costa-17 had no PIP on board and is therefore limited to 1000 µm
Notable is that the Costa-17 PSDs include not only Arctic, but also mid-latitudes and tropical clouds. This means that the observed structures in the PSDs depend not on geography but on temperature and environmental conditions.

[Figure]

Moser et al.(2025)

Costa et al. (2017)

[Figure]

| | PSD | | Aspherical fraction for Dp 20–50 μm | Particles Dp > 50 μm | Dominant mass mode |
|---|---|---|---|---|---|
| Mostly liquid | Type 1 | a | Zero | Drizzle drops/few ice crystals possible | Dp < 50 μm |
| Coexistence | | b | Low | Glaciated | Dp < 50 μm |
| Secondary ice | | c | High | Glaciated | Dp < 50 μm |
| Large ice/ WBF | Type 2 | | High | Glaciated | Dp > 50 μm |

**Figure 1**

Particle size distributions (PSDs) of mixed- phase clouds, measured by Moser-25 (upper panel) and Costa-17 (middle panels). The bottom panel shows the mixed-phase cloud type classification of Costa-17 (their Table 15).

Moser-25 and Costa-17 both found two mixed-phase cloud types:

Type 1: MPC | 'Coexistence'    (Moser-25 | Costa-17),
Type 2: MPH | 'Large ice/WBF' (Moser-25 | Costa-17).

The PSDs are very similar, which can be seen when comparing the blue (Type 1) and green (Type 2) horizontal lines, drawn to guide the eye.

Instruments used for the measurements: Costa-17: CAS-DPol, CIP; Moser-25: CDP, CIP, PIP; all instruments were manufactured by DMT, Boulder, CO, USA. For more information seey Costa-17 and Moser-25.

The interpretation of PSDs of *Type 1* (MPC | 'Coexistence' clouds; Moser-25 | Costa-17), is that:

– the many small ($< 50$ µm) cloud particles are liquid droplets and the larger ones are ice crystals. Moser-25 proves this with measurements that show saturation/supersaturation with respect to water of the ambient air; Costa-17 explained this in the same way, but could not prove it due to a lack of humidity measurements. Instead, they used depolarization measurements in this size range to show that the cloud particles were spherical (see the Table at the bottom of Figure 1: the aspherical fraction of the small cloud particles is low for 'Coexistence' clouds).

– for the larger cloud particles, Moser-25 assumes that they are frozen based on their size; Costa-17 shows this by determining the asphericity of the cloud particles.

The PSDs of *Type 2* (MPH | 'Large ice/WBF' clouds; Moser-25 | Costa-17) are interpreted as completely glaciated mixed-phase clouds in both studies. These clouds are found in ambient air that is subsaturated with respect to water and saturated/supersaturated with respect to ice. All small liquid droplets have evaporated, and the larger ice crystals have grown due to the released water vapour - this is called the Wegener-Bergeron-Findeisen (WBF) process.

However, in both studies, particles are still present in the size range $< 50$ µm, though Costa-17 found these small particles up to $\sim 20$ µm, while Moser-25 the did not detect particles larger than 6 µm. This may be due to the smaller data base of Moser-25 (Costa-17: 38.6 h, Moser-25: 19.4 h). Costa-17 reported that PSDs without small particles where detetcted, but not frequently. Note that these particles are present at similar concentrations across all temperature ranges in the Costa-17 dataset.

Moser-25 identified these particles as dissolved sea salt particles. Costa-17 did not explain the occurrence of these particles, but showed that they contain a high fraction of aspherical particles. The measurement of the asphericity of the particles fits very well with the results of the new study.

**Summary of the comparison**

Overall, Moser-25's study confirms Costa-17's findings for the two types of mixed-phase Arctic low level clouds (between 255 and 265 K).

Since Moser-25 investigate Arctic mixed-phase clouds and Costa-17 a mixture of Arctic, mid-latitude, and tropical clouds, the similarity of the PSDs shows that their structures depend not on geography but on temperature and environmental conditions. The dependence on temperature means that clouds are found at different altitudes, lowest in the Arctic and highest in the tropics.

A new result of Moser-25 is that the smaller particles ($< 6$ µm) present in mixed-phase clouds that are completely glaciated by the WBF process (*Type 2*) are dissolved sea salt particles (haze particles). This result is obtained by Moser-25 and confirmed by Costa-17's aspherity measurements, but the Costa-17 PSDs range up to $\sim$20 µm.

**Recommendations**

1. To contextualize the study within the broader landscape of existing research, I recommend to include the comparison with Costa-17 into the manuscript, in particular to point out

   **a**     the similarity of the PSDs found by Moser-25 and Costa-17,

   **a-1** indicating that the structures of the PSDs depend not on geography but on temperature and environmental conditions.

   **b**     that the finding that the small mode particles ($< 6$ µm) are dissolved sea salt particles is confirmed by the asphericity measurements of Costa-17, and to discuss that Cost-17 found small aspherical particles up to $\sim$20 µm.

2. I also recommend reconsidering the names of the mixed-phase cloud types.

   Often, the -well known- two types of mixed-phase clouds are summarized under the acronym MPC. In Moser-25, MPC only includes the cloud type in which small liquid droplets coexist with large ice crystals.

   For the completely glaciated mixed-phase clouds the, term MPH (mixed-phase haze) is introduced. This is somewhat misleading because 'haze' (in particular Arctic haze) refers commonly to a size spectrum of grown aerosol particles, not to a cloud. However, this type of completely glaciated cloud is usually considered as glaciated mixed-phase cloud with a very low concentration of large ice crystals, which, however, represent the dominant mass mode (see Figure 1, bottom panel) .

   Therefore, I would recommend names that make it clear that both types are clouds, maybe MPC_coex for the Type 1 and MPC_haze for the second?

With kind regards,     Martina Krämer
Johannes Gutenberg-University Mainz, Germany

**Reference:**

- **Costa, A.**, Meyer, J., Afchine, A., Luebke, A., Günther, G., Dorsey, J. R., Gallagher, M. W., Ehrlich, A., Wendisch, M., Baumgardner, D., Wex, H., and Krämer, M.: Classification of Arctic, midlatitude and tropical clouds in the mixed-phase temperature regime, Atmos. Chem. Phys., 17, 12219–12238, https://doi.org/10.5194/acp-17-12219-2017, 2017.

---

## Author Comment (AC1)

Review of *"The Arctic Low-Level Mixed-Phase Haze Regime and its Microphysical Differences to Mixed-Phase Clouds"*, by Manuel Moser, Christiane Voigt, Oliver Eppers, Johannes Lucke, Elena De La Torre Castro, Johanna Mayer, Regis Dupuy, Guillaume Mioche, Olivier Jourdan, Hans-Christian Clemen, Johannes Schneider, Philipp Joppe, Stephan Mertes, Bruno Wetzel, Stephan Borrmann, Marcus Klingebiel, Mario Mech, Christof Lüpkes, Susanne Crewell, André Ehrlich, Andreas Herber, and Manfred Wendisch, egusphere-2025-3876.

**Response to reviewer 1**

Dear reviewer,
We are very grateful for your valuable feedback and suggestions which helped us to improve the manuscript. The manuscript has been thoroughly revised and point-by-point responses have been prepared. Please find below our replies, highlighted in blue, along with changes made in the manuscript, highlighted in orange. The revised manuscript is also provided with tracked-changes for clarity.

**Major comments**
These are comments regarding the comprehension of the manuscript and include questions to help the clarification of the findings. The authors should consider addressing them.

- Line 45: The authors hint at the potential of MPH for the atmospheric radiation budget. However, throughout the manuscript no evidence is shown for that, and in the end, the authors state that MPH are optically thin (Line 394). I am wondering what is their importance? This also connects to how often this regime was identified, and especially with no liquid layer at cloud top. Because for MPC the liquid layer is the driver for the cloud top cooling and has the biggest impact of the radiative budget. Can you please elaborate on that more?

We thank the reviewer for this valuable comment. Indeed, this study does not include a quantitative calculation or direct measurement of the optical thickness. The statement that mixed-phase haze (MPH) is optically thinner than mixed-phase clouds (MPC) is based on visual impressions from synchronized onboard photographs and video footage, which were correlated with the respective cloud regimes.
In the introduction, we assume that the MPH regime is optically thin. Therefore, we highlight the general relevance of optically thin clouds in the context of sensitivity, as even small changes in their microphysical properties can lead to substantial differences in their radiative effects, e.g. downward thermal-infrared emission warming the surface. However, we do not claim that the MPH regime itself has a large impact on the radiative budget.
To provide a more objective comparison, we now omit the optical thickness statement from the manuscript and instead use the extinction coefficient to quantify and compare the optical properties of MPH and MPC. Since the data from the Polar Nephelometer (PN) are not available in sufficient quality for the HALO-(AC)³ campaign, we refer to PN data from the AFLUX campaign presented in Fig. 8 in Moser et al. (2023b). In this figure extinction coefficients for the individual cloud regimes are shown. For the specific comparison between MPH and MPC, these data are summarized in R_Fig. 1. From these data, the median [25th/75th percentile] extinction coefficient is 0.7 km$^{-1}$ [0.2 / 1.8 km$^{-1}$] for the MPH regime (regime 2a + 2b) and 5.2 km$^{-1}$ [2.6 / 10.1 km$^{-1}$] for the MPC regime

(regime 2c). This confirms that the extinction coefficient of MPH is significantly lower than that of MPC, reflecting its optically thin nature.

[Figure]

R_Figure 1: Frequency distribution of the extinction coefficient (Ext. Coef.) measured with the Polar Nephelometer for the (a) mixed-phase haze (MPH; 2a +2b) and (b) mixed-phase cloud (MPC; 2c) regimes. The data were obtained during the Arctic field campaign AFLUX in spring 2019. Original figure is presented in Moser et al. (2023), Figure 8.

Changes made in the manuscript:
- We have changed the sentence in line 391 to: "Consistent with the visual observations, the extinction coefficient of MPH is significantly lower than that of MPC, reflecting their optically thinner property. Values for the extinction coefficients are provided in Fig. 8 of Moser et al. (2023b). Based on the measurements during spring 2019, the median [25th/75th percentile] extinction coefficient is 0.7 km$^{-1}$ [0.2 km$^{-1}$ / 1.8 km$^{-1}$] for the MPH regime (2a + 2b) and 5.2 km$^{-1}$ [2.6 km$^{-1}$ / 10.1 km$^{-1}$] for the MPC regime (2c)."
- Sentence in line 405 changed to: "This difference is also reflected in the substantially reduced extinction coefficient (Ext. Coef.) observed for MPH."
- Figure 8: Optical thickness ($\tau$) was changes to the extinction coefficient (Ext. Coef.)

**Minor comments**
These are editorial comments helping to improve the formatting and readability of the manuscript. The authors should consider adding them.
- Line 13: While reading I was wondering if the MPH is a state instead of MPCs, but it appears that it is most often found beneath the MPC, which is stated here. However, there are cases of MPH without a MPC identified. I think the abstract would benefit from a clear statement on the occurrence and persistence of MPH.
  Reply: We thank the reviewer for the suggestion to emphasize the relevance of MPH by highlighting its frequent occurrence. We have extended the sentence in Line 4 of the abstract with: „, with MPH observed about eight times more frequently than MPC.". This statement is based on the results shown in Fig. 8, indicating a 4 % fraction of MPC and a 34 % fraction of MPH.

- Line 21: "Many factors influencing Arctic amplification are discussed (Wendisch et al., 2023a)", is an incomplete sentence. Either you say "discussed in" or you rephrase it in such a way that it is clear that a list of possible reasons is following.

Reply: We have changed the sentence to: "Many factors influencing Arctic amplification are discussed in the literature (Wendisch et al., 2023a)"

- Line 28: That MPCs are in a quasi-steady state is true, however, the reason is not only dynamical processes but rather a superposition of dynamics, turbulence, radiation, and cloud microphysics. This should be clearly stated, as it is still a challenge to fully understand how these clouds remain persistent given their metastable thermodynamic state.
  Reply: We have changed the sentence to: "These clouds persist in a quasi-steady state due to a complex interplay of dynamics, thermodynamic structure, radiation, and microphysics, which interact to maintain the liquid phase in spite of its metastable thermodynamic state (Morrison et al., 2011)."

- Line 36: replace "including those" by (e.g., ....).
  Adopted

- Line 44: using a threshold of LWC = 0.01 g m→3
  Adopted

- Line 48: mixed-phase temperature regime needs to be defined (in numbers)
  Reply: We have changed the sentence to: "Costa et al. (2017) observed a low number concentration (N < 1 cm−3) in some cloud types when analysing a large in-situ data set of clouds in the mixed-phase temperature range between 0 °C and -38 °C."

- Line 49: the Wegener-Bergeron-Findeisen process should be at least explain in 1-2 sentences, because it appears throughout the manuscript.
  Reply: We assume that the Wegener-Bergeron-Findeisen process is familiar to readers working in this field and therefore do not provide a detailed explanation in the manuscript. However, we have added two relevant references (Pruppacher and Klett, 2010; Storelvmo and Tan, 2015) at the first occurrence of the term. Please also see our response to Reviewer 2 regarding the comment "Line 49: Add citation for the Wegener–Bergeron–Findeisen (WBF) process."

- Line 50: One-sentence paragraphs should be avoided.
  Reply: Thank you for the comment. Following a suggestion from Reviewer 2, this paragraph has been extended and is no longer a one-sentence paragraph.

- Introduction: While the abbreviations MPC (also it should be MPCs if it is used in the plural form) and MPH are introduced in the abstract, they are only defined later in the manuscript text, even though the regimes are discussed already in the introduction. Please double-check that you introduce the abbreviations at the earliest appearance and then use them throughout the text. This goes for several abbreviations, such as LWC and IWC as well.

  Reply:
  We have carefully reviewed the use and introduction of all abbreviations throughout the manuscript. The terms mixed-phase cloud (MPC) and mixed-phase haze (MPH) are now introduced at their first occurrence in the main text and consistently used

thereafter. In the abstract, both terms remain written out in full to ensure that it can be read independently from the manuscript.
The abbreviations MPC and MPH are only used where the respective regime is explicitly meant.
In addition, the abbreviations LWC, IWC, CWC, N, $D_{eff}$, ABL, MIZ, $RH_w$, $RH_{ice}$, CCN, INPs, PSD, and SIC are now correctly defined and used consistently throughout the paper.

- Line 66: what does "partially close proximity" mean? Can you just quantify it as you do it later in the text?
  Reply: We cannot provide a more precise quantification here, as the flight activities conducted with the FAAM and ATR aircraft were not part of this study. They are mentioned explicitly to acknowledge their coordinated operation within the HALO-$(AC)^3$ framework and to draw attention to related measurement activities.

- Line 72: "had" to has
  Adopted

- Line 74: reference to Figure 1 when you introduce the location
  Adopted, we have added "(see Fig. 1)" at the end of the sentence.

- Line 84: CWC needs to be introduced, especially since I assumed you meant LWC. It should be clear that here the total water content (TWC) is meant. Also a short sentence on the threshold should be done, and not only referring to a previous study.
  Reply: We have changed the sentence in line 84 to: "The microphysical low-level cloud dataset from Polar 6 consists of a total of 19.4 h (< 1000 m, and cloud threshold CWC > 2×10−4 g m−3 according to Moser et al., 2023b; CWC = LWC + IWC), collected during 13 flights in March and April 2022." And deleted the sentence in line 243: "The CWC is calculated as the sum of LWC and IWC."

- Line 93: "N" needs to be defined
  Reply: N is now defined before that line

- Line 121: Why are legs from horizontal flights less significant?
  Reply: We assume that the reviewer's question is "Why are legs from vertical flights less significant?". Vertical flight legs are less statistically significant than horizontal ones because environmental and thus microphysical properties typically vary more rapidly in the vertical than in the horizontal direction. During horizontal legs, data can be averaged over longer distances, which increases the statistical robustness of the measurements. In contrast, vertical profiles are evaluated at 1 Hz resolution, whereas in studies focusing on horizontal in-situ cloud data, averaging is often performed over several minutes.
  In addition, the airflow conditions around the in-situ cloud probes can change during ascents and descents, which are less well characterized than during steady horizontal flight legs.

- Line 126: Suggestion for reformulation: Moser et al. (2023b) introduced a method for determining ...
  Adopted

- Line 131: The sentence regarding the thresholds is somewhat confusing, and should be rephrased, such that word repetition is avoided.
  Sentence changed to: Consequently, when applying the same procedure as in Moser et al. (2023), the resulting threshold values separating the individual regimes differ slightly.

- Table 1: the abbreviation were not introduced.
  Reply: This is now corrected in the revised version.

- Line 142: given that CAO and WAI are only used twice, consider not using the abbreviations and just write it out.
  Adopted

- Line 152: The classification ocean/land I find confusing as it should indicate that the air mass was mostly influenced by the ocean. In Line 182 the phrasing is also slightly different for the ocean/land mask, which makes it more confusing.
  Reply: We agree with the reviewer. One option would be to name the mask simply "ocean" and mention in one sentence that a minor influence from land cannot be excluded. However, we prefer to remain transparent and scientifically precise. Therefore, we chose to describe the mask as "ocean/land" to accurately reflect the possible influence of both surface types.

- Line 165: which data were now used?
  Reply: Only the meteorological data from the dropsondes were used, as the nose boom data provided reliable measurements on only a few days. For the days when both datasets were available, the correlation approach between the Polar 6 in-situ cloud measurements and the dropsonde data was verified. Since this validation is not part of the manuscript, the corresponding sentences were removed to avoid confusion:
  ", which limited the availability of reliable data to only a few days. However, these data were used to validate the correlation approach between Polar~6 and the dropsonde data."

- Line 168: the information in the brackets for SIC can be omitted as it was defined before
  Adopted

- Line 170: Suggestion for reformulation: ..., at which the minimum temperature occurs (i.e., minimum of temperature inversion).
  Adopted

- Figure 2: For all figures the colors should be revised such that they are color-vision-deficiency friendly (especially avoiding red and green lines). I was wondering, if the

grey dropsonde profiles can be stratified by the origin as it was done for the mean value. You could use just a lighter color of the mean value to indicate that. Moreover, asterisks could be used to mark the mean ABL height.

Reply: All figures have been rechecked and, where necessary, adjusted to be color-vision-deficiency friendly. In Fig. 2, the dropsondes are now categorized according to their release location, over the sea ice, over ocean, and in the MIZ. In addition, the color scheme (consistent with Fig. 3) now distinguishes the air mass origin.

- Line 172: First the figure should be introduced and then the interpretation should be done. Move the first sentence to after the sentence ending on Fig. 3.
  Adopted

- Line 177: The sentence "Consequently, ..." is a repetition of what was said in the sentence before. Can this be consolidated and maybe combined?
  Reply: We acknowledge that the sentence partly repeats the previous statement; however, it also provides a simplified explanation that improves readability and emphasizes the key message. Therefore, we decided to keep it as it is.

- Line 184: I do not see how Fig. 3 informs us on the vertical structure of the ABL? Do you mean the vertical extent? You only investigate the structure later in the manuscript.
  Reply: We have changed "vertical structure" to "vertical extend"

- Line 187: Is it a surprise that the large-scale conditions define the temperature, but local processes determine the boundary layer dynamics? If so, this should be emphasized. If not, I would suggest to add references to contextualize your statement.
  This statement specifically refers to the air masses crossing the sea ice edge, e.g., warm air masses moving northwards form ocean to sea ice and cold air masses moving southwards from sea ice to open ocean. In these cases, the present air mass is mostly driven by synoptic processes. However, air mass transformation is triggered by the changes surface and this mostly affects the ABL. E.g., in cold air outbreaks, the temperature at top of the ABL almost remains constant, while the strong surface fluxes over the ocean lead to a mixing of the lower atmosphere that causes an increase of the ABL height (Wendisch et al., 2025).
  The citation (Wendisch et al., 2025) was added in line 188.

- Line 189: The definition of the normalized altitude should be introduced when it is actually used in Section 3.2.
  Reply: We understand the reviewer's point. However, the definition is placed in this section because it is part of the methodology and therefore fits best within Section 2: Methods.

- Line 203: "as" is missing and the explanation for combining the regimes for MPH could be supported by an appendix figure, if the authors want to.
  Reply: The word "as" has been added. We appreciate the suggestion to include an additional figure in the Appendix. However, we believe that the description in the text

is sufficiently clear, and further details on the microphysical properties of the individual regimes 2a and 2b are already presented in Moser et al. (2023).

- Line 220: The introduced regimes and abbreviations should be used explicitly and not converted back to full text (whole paragraph).
  Adopted

- Figure 4: add in the caption that the percentiles were calculated based on bootstrapping.
  We have added to the figure caption: "The percentiles were calculated based on bootstrapping."

- Line 244: Why the reference to Moser et al., 2023b: It is just the sum of IWC and LWC, no? Here the CWC defintion needs to come earlier in the manuscript.
  Reply: This issue has been resolved during the revision process in response to a previous comment.

- Line 255: The Wegener-Bergeron-Findeisen process describes in principle the evaporation of cloud droplets and the growth of ice crystals. However, for cloud droplets to evaporate, water subsaturated conditions must exist, otherwise they are not evaporating. Then both cloud droplets and ice crystals would grow. So, the WBF process by itself does not imply water saturation, but rather the available cloud droplets. So one needs to be careful when arguing with the WBF process. Also I would argue that the WBF process is not persistent.
  Reply: We agree with the reviewer that the original sentence was not scientifically accurate. We have therefore revised the sentence in Line 255 as follows: "In a persistent mixed-phase cloud state, a quasi-steady balance exists between the ongoing Wegener-Bergeron-Findeisen process, which transfers water mass from liquid droplets to ice crystals, and dynamical processes that compensate the resulting mass transfer."

- Line 257: the word cloud is missing, but anyway the abbreviation MPC should be used. I agree that there is water saturation, but the WBF process is definitely not the only process helping to sustain that. This is too-short handed argued.
  Reply: We have added the term mixed-phase state at this point. We intentionally avoid using the abbreviation MPC here, as throughout the manuscript this abbreviation is only used when explicitly referring to the in-situ identified mixed-phase cloud regime.

- Line 260: What do you mean by greater sensitivity of the liquid phase to the enviroment?
  Reply: By this statement we refer to the higher sensitivity of liquid particles to environmental relative humidity compared to ice particles. Because of the short relaxation time of liquid droplets (only a few seconds; Korolev et al., 2017), the relative humidity distribution within mixed-phase clouds is mainly determined by the liquid phase. In contrast, ice crystals can persist in subsaturated air masses due to a much longer phase relaxation time, which is comparable to the lifetime of an entire ice cloud (Krämer et al., 2009; Rollins et al., 2016; Korolev et al., 2017).

- Line 262: higher supersaturation than what?
We have adopted this sentence: "However, liquid clouds are observed more frequently at colder temperatures, resulting in slightly higher levels of supersaturation with respect to ice than is observed within MPCs."

- Line 265: Isn't it the defitnio of MPH and ice that RH < 100 %?
Reply: This refers to an observation in our dataset, which we attribute to the mixed-phase haze (MPH) regime.

- Line 274: Are these really stable thermodynamic conditions? Wouldn't the existence of ice crystals with the haze droplets mean some metastable state?
Reply: We thank the reviewer for this comment. Given the frequent occurrence of these cloud regimes, we assume that they represent a relatively stable or metastable condition. However, we cannot demonstrate that this stability is of thermodynamic origin. To avoid overinterpretation, we have therefore removed the word thermodynamic from the sentence: "…, suggesting the presence of a stable condition."

- Line 314: Can you explain the process how GCCN are impacting ice sublmation and cloud droplet growth?
Reply: The study by Ji et al. (2025) shows that giant CCN can reduce the water vapor pressure over large solution droplets, which may shift the vapor equilibrium between liquid and ice. This can suppress ice growth or enhance droplet growth, thereby affecting the balance between ice sublimation and droplet condensation. We have added the sentence: "Large CCN can lower the equilibrium vapor pressure over the droplet surface, potentially inhibiting ice growth and extending the mixed-phase lifetime."

- Line 323: Why did you choose 300 m?
Reply: Since this analysis also investigates the influence of the surface on the haze particles, it was important to consider only data within the ABL. The altitude of 300 m was chosen as a compromise: low enough to ensure that most data points are within the ABL, but high enough to include a sufficiently large statistical sample.

- Line 327: dotted should be dashed
Adopted

- Line 336: I thought the MPH have water subsaturated conditions, how can there be liquid water present?
Reply: Haze droplets consist of a mixture of water and dissolved salts. These particles are not activated cloud droplets but exist in equilibrium with the subsaturated environment. Although they are much smaller than typical cloud droplets, they still contain a substantial fraction of liquid water.

Figure 7: The colors could be in agreement with Figure 3 and instead of using the same linestyle for separation the total, use maybe dotted.
Adopted

- Figure 8: Can you turn the pie chart in such a way that "0 %" starts at the top of the circle? This way it is more intuititve.
  Adopted

- Figure 9: What is the white shading in a, c, and e?
  Reply: We have added the following sentence in the figure caption: "Insufficient data due to low statistics are grayed out."

- Line 376: Why is the glaciation efficiency higher in more stable conditions?
  Reply: We thank the reviewer for pointing this out. We agree that the original sentence was misleading, and we have removed it in the revised manuscript.

- Line 385: Here an assessment of the lifetime of MPH would be great.
  Reply: We agree with the reviewer that an assessment of the lifetime of MPH would be highly interesting. However, with the available in-situ aircraft data, it is only possible to estimate this to a limited extent. A detailed analysis would require dedicated flight patterns specifically designed to follow the temporal evolution of a mixed-phase haze layer.
  Nevertheless, one goal of this study is to motivate future measurements and complementary observations of the MPH regime using other techniques. For example, such investigations could be carried out using airborne or ground-based lidar observations to identify potential MPH signatures, observe their evolution, and estimate their lifetime. Moreover, existing lidar or remote-sensing datasets might already contain indications of mixed-phase haze layers that were previously not classified as clouds due to their low extinction coefficients. Revisiting such datasets with the findings from this study in mind could provide valuable insights before new dedicated measurements are conducted. Such an analysis would go beyond the scope of this paper. Our conclusion regarding the potentially long lifetime of the MPH regime is therefore based on its frequent occurrence and on the finding that the MPH regime represents a equilibrium state with the environment due to its microphysical composition.

- Line 399: Does not the mixed-phase state imply that we have liquid and ice crystals? So this is circular?
  Reply: We would like to clarify the wording. The term mixed-phase state refers to the regimes 2a, 2b, and 2c, all of which contain both liquid water and ice. However, in regimes 2a and 2b the liquid phase is present in haze particles, whereas in regime 2c it is found in cloud droplets.

- Conclusion point 2 and 4: These are the same to me and could be combined.
  Reply: In conclusion point 2, the microphysical composition of the two regimes is described, whereas conclusion point 4 focuses on the environmental conditions under which the two regimes are observed and the corresponding physical explanation. To improve readability, we have swapped points 3 and 4 in the revised version.

- Line 406: The optical thickness was only assessed visually, no? Can you quantify it?

Reply: This comment has been addressed during the revision process as part of the major comment. We thank the reviewer for this helpful remark.

- Line 421: Are MPH really persistent? You did not show anything regarding lifetime?
  Reply: See response to the comment on Line 385.

References:

*Moser, M., Voigt, C., Jurkat-Witschas, T., Hahn, V., Mioche, G., Jourdan, O., Dupuy, R., Gourbeyre, C., Schwarzenboeck, A., Lucke, J., Boose, Y., Mech, M., Borrmann, S., Ehrlich, A., Herber, A., Lüpkes, C., and Wendisch, M.: Microphysical and thermodynamic phase analyses of Arctic low-level clouds measured above the sea ice and the open ocean in spring and summer, Atmospheric Chemistry and Physics, 23, 7257–7280, https://doi.org/10.5194/acp-23-7257-2023, 2023b.*

*Korolev, A., McFarquhar, G., Field, P. R., Franklin, C., Lawson, R. P., Wang, Z., Williams, E., Abel, S. J., Axisa, D., Borrmann, S., Crosier, J., Fugal, J., Krämer, M., Lohmann, U., Schlenczek, O., Schnaiter, M., and Wendisch, M. (2017). "Mixed-Phase Clouds: Progress and Challenges". In: Meteorological Monographs 58, pp. 5.1–5.50. DOI: 10.1175/amsmonographsd-17-0001.1.*

*Krämer, M., Schiller, C., Afchine, A., Bauer, R., Gensch, I., Mangold, A., Schlicht, S., Spelten, N., Sitnikov, N., Borrmann, S., Reus, M. de, and Spichtinger, P. (2009). "Ice supersaturations and cirrus cloud crystal numbers". In: Atmospheric Chemistry and Physics 9.11, pp. 3505–3522. DOI: 10.5194/acp-9-3505-2009.*

*Rollins, A. W., Thornberry, T. D., Gao, R. S., Woods, S., Lawson, R. P., Bui, T. P., Jensen, E. J., and Fahey, D. W. (2016). "Observational constraints on the efficiency of dehydration mechanisms in the tropical tropopause layer". In: Geophysical Research Letters 43.6, pp. 2912–2918. DOI: 10.1002/2016gl067972*

*Wendisch, M., Kirbus, B., Ori, D., Shupe, M. D., Crewell, S., Sodemann, H., and Schemann, V.: Observed and modeled Arctic airmass transformations during warm air intrusions and cold air outbreaks, Atmospheric Chemistry and Physics, 25, 15 047–15 076, https://doi.org/10.5194/acp-25-15047-2025, 2025.*

---

## Author Comment (AC2)

Review of *"The Arctic Low-Level Mixed-Phase Haze Regime and its Microphysical Differences to Mixed-Phase Clouds"*, by Manuel Moser, Christiane Voigt, Oliver Eppers, Johannes Lucke, Elena De La Torre Castro, Johanna Mayer, Regis Dupuy, Guillaume Mioche, Olivier Jourdan, Hans-Christian Clemen, Johannes Schneider, Philipp Joppe, Stephan Mertes, Bruno Wetzel, Stephan Borrmann, Marcus Klingebiel, Mario Mech, Christof Lüpkes, Susanne Crewell, André Ehrlich, Andreas Herber, and Manfred Wendisch, egusphere-2025-3876.

**Response to reviewer 2**

Dear reviewer,
We are very grateful for your valuable feedback and suggestions which helped us to improve the manuscript. The manuscript has been thoroughly revised and point-by-point responses have been prepared. Please find below our replies, highlighted in blue, along with changes made in the manuscript, highlighted in orange. The revised manuscript is also provided with tracked-changes for clarity.

Major comments

**1: Clustering and Physical Interpretation**
The authors employ clustering and statistical analysis to interpret the underlying mechanisms of the observed phenomena. However, observational data alone do not directly confirm physical processes (e.g., during HALO-(AC)³, the synoptic situation primarily controls the ABL top temperature, while surface-driven processes determine its vertical extent). The logical connections could be strengthened by incorporating back-trajectory analyses—especially in Section 2.3, and at Lines 183 and 207 in Section 3.1.1.

Reply: We thank the reviewer for this comment. It is not entirely clear whether the reviewer refers to the clustering of meteorological parameters shown in Figure 3, or the clustering of microphysical parameters distinguishing regimes 2a and 2b in Section 3.1.1.
In both cases, backward trajectory analyses were already applied to interpret the differences between the identified clusters.
For the meteorological classification in Figure 3b, the color coding is directly based on 24-h backward trajectories, as described in Line 143: "Backward trajectory analyses were conducted to determine the dominant surface type over which the low-level air masses had resided during the 24 h prior to their in-situ measurement by Polar 6."
Hence, each meteorological cluster in Fig. 3b implicitly contains the information on air-mass origin and synoptic conditions.
For the microphysical regimes 2a and 2b, the hypothesis that their differences are primarily driven by surface type and corresponding air-mass history is discussed in detail in Moser et al. (2023b), which also relied on backward trajectories to support this interpretation.
If the reviewer referred to a different aspect of clustering, we would be happy to clarify this further in a revised version of the manuscript.

**2: Clustering Justification and Consistency**

**(i):** In Table 1, MPH is divided into sub-clusters 2a and 2b, which are later recombined for microphysical comparisons with MPC. The rationale for creating sub-clusters (e.g., 2a vs. 2b, or 1a vs. 1c) is unclear.

Line 136 mentions that Table 1 lists the thresholds defining each regime, but the basis for these choices isn't fully explained.

**(ii):** Additionally, there's no figure provided to support Lines 288–289.

**(iii):** It appears that in many sections, sub-clusters are merged (e.g., 2a+2b → 2), raising the question of whether the initial subdivision is necessary. While Lines 200–221 are logically structured, please clarify the importance of including these sub-clusters in the context of the central narrative. Why are these regime details crucial for the main scientific conclusions?

Reply:

We thank the reviewer for raising these constructive questions. They can be divided into three main aspects, addressed below as (i)–(iii).

(i) and (iii)

We appreciate the reviewer's remark that the physical reasoning behind the applied classification was not sufficiently clear in the manuscript. The classification scheme used in this study builds directly on the work of Moser et al., 2023b, where a clustering of all AFLUX and MOSAiC-ACA data revealed seven local maxima in the N-Deff space. These maxima define the regimes 1a, 1b, 2a, 2b, 2c, 3, and 4.

The same cluster structures are reproduced in the HALO-(AC)3 dataset, which allows the regimes in the current study to be assigned consistently to those defined in Moser et al., 2023b. The boundaries were slightly adjusted to match the maxima of the new dataset, but the classification algorithm remains identical.

The naming convention (1a - 4) follows the microphysical regime type: ice (1a, 1b), mixed-phase (2a, 2b, 2c), liquid (3), and aerosol (4). In the present study, which extends the work of Moser et al., 2023b, the mixed-phase regime (2a, 2b, 2c) is reexamined in greater detail. The results show that a further differentiation within this regime is physically meaningful, specifically between mixed-phase haze (2a, 2b) and mixed-phase cloud (2c). A new naming scheme was intentionally avoided to maintain consistency with the previous study.

We acknowledge that the basis for this decision was not explained clearly enough in the original version and have therefore added the following clarification at the end of Section 2.2 (Line 139):

"Please note that the classification into regimes 1a, 1b, 2a, 2b, 2c, 3, and 4 originates from Moser et al. (2023b). The present study builds upon these findings and focuses specifically on the mixed-phase regime. The results suggest that refining this classification would be appropriate, as the mixed-phase regime can be subdivided further based on its distinct microphysical characteristics. However, to ensure consistency with the previous work and to avoid unnecessary complexity, the original nomenclature is retained. Accordingly, regimes 1a and 1b (hereafter referred to as the ice regime) represent ice clouds, regimes 2a and 2b (the MPH regime) represent

the mixed-phase haze conditions, regime 2c (the MPC regime) represents classic mixed-phase clouds, regime 3 (the liquid regime) represents liquid clouds, and regime 4 (the aerosol regime) corresponds to aerosol measurements."

(ii):
We believe that the information supporting Lines 288-289 is already clearly presented in the manuscript. The haze droplet sizes (3-6 μm) are shown and discussed in Section 3.1.1. The corresponding dry diameters (1-3 μm) were derived from the hygroscopic growth factor (median = 2.2; 25th/75th percentiles = 2.1/2.6) using the relations 3μm/2.6≈1μm  and 6μm/2.1≈3μm.
The statement that the observed droplets exist above the deliquescence point of NaCl is evident in Figure 5b. Therefore, no additional figure is required. To make this more transparent, we replaced the sentence at Line 288 with the following:
"Based on this hygroscopic growth factor and the observed haze droplet sizes ranging from 3 μm to 6μm, the estimated dry diameters of the solute particles are between 1μm and 3μm, calculated from the observed wet diameter range (3–6μm) divided by the 25th and 75th percentile of the hygroscopic growth factor."

**3: Suggestions for Figure 2**
Figure 2 shows large variability, and the gray lines do not provide as much value as intended. I suggest replotting the three profile types in separate panels with a shared axis range, using the color scheme from Figure 3. Each panel can include all dropsondes of that type along with an averaged profile. This restructuring would facilitate cross-comparison with Figure 3, support the discussion on "exceptions and normals" (p. 8), and better illustrate inversion frequency.

Reply: Figure 2 has been revised accordingly. The three profile types are now displayed in separate panels with a consistent axis range, following the color scheme used in Figure 3. This restructuring enables a direct comparison between Fig. 2 and 3 and allows the reader to easily identify the air mass origin and surface type for each individual temperature profile.
Adapted figure caption: Temperature profiles measured by dropsondes from HALO and Polar 5. The data are used for the analysis in Sect. 3.1.2 and Sect. 3.2. The dropsondes are separated by the underlying surface conditions: (a) over sea ice, (b) within the marginal sea ice zone (MIZ), and (c) over the open ocean. The color of each individual dropsonde represents the air mass origin. For each surface condition, an averaged temperature profile is additionally shown. All dropsondes used in this figure are listed in Table A1.

We have added the following in line 167:
The color of each individual dropsonde indicate the air mass origin. Additionally, the dropsondes were classified based on satellite data to determine whether they were deployed over sea ice (Fig. 2 (a); SIC > 80 %), the open ocean (Fig. 2 (c); SIC < 20 %), or the MIZ (Fig. 2 (b); 20 % ≤ SIC ≤ 80 %).

**4: Further Analysis of Figure 4 by Environment Type**

Figure 4 is already informative, but the analysis could be enhanced by breaking it down by environment—such as marginal ice zone (MIZ), open ocean, and sea ice. Including additional subplots by surface type would provide valuable insights for readers and future studies.

Reply: We thank the reviewer for this valuable suggestion to further enhance the information content of Figure 4. However, the intention of this figure is to explicitly highlight the microphysical differences between the mixed-phase haze (MPH; regimes 2a+2b) and the mixed-phase cloud (MPC; regime 2c), emphasizing the necessity of distinguishing these two regimes.

A subdivision by surface type (sea ice, MIZ, open ocean) would go beyond the scope and purpose of this particular figure and cannot be meaningfully implemented with the available dataset. In contrast to AFLUX (see Moser et al., 2023b), only very few MPCs were observed over sea ice during the HALO-(AC)[3] campaign (see R_Fig. 1). Therefore, a statistically robust comparison of PSDs by surface type is not feasible.

Nevertheless, R_Fig. 2 illustrates the particle size distributions of regimes 2a, 2b, and 2c. The largest differences between regimes 2a and 2b occur in the ice-crystal size range. Since regime 2a is more frequently observed over sea ice and regime 2b over open ocean, these differences can be attributed to surface influence, consistent with the findings of Moser et al. (2023b). Moreover, detailed PSDs of these regimes are already published in Fig. 7 in Moser et al. (2023b).

For clarity and focus, Fig. 4 in the current manuscript is therefore kept in its present form, as it aims to illustrate specifically the microphysical distinction between MPH and MPC.

[Figure]

R_Figure 1: Frequency of occurrence for each particle regime (1a, 1b: Ice particles; 2a, 2b, 2c: Mixed-phase particles; 3: Liquid particles; 4: Aerosol particles), separated by surface conditions for the HALO-(AC)[3] low-level cloud data (< 1000 m). The values are normalized by the respective surface condition. This figure is taken from the PhD thesis Moser (2024) (Fig. 55 in http://doi.org/10.25358/openscience-11192).

[Figure]

R_Figure 2: Particle size distribution of classic mixed-phase cloud (2c) and the mixed-phase haze sub-regimes 2a and 2b. The PSD lines give the median value, calculated the same way as stated in the manuscript.

**5: Clarity in Figure 5**

Figure 5 is the most difficult to interpret due to the complexity of the color-coded "step" histograms. Would it be possible to separate Ice and MPH into an additional column or panel to reduce visual clutter?

Reply: We agree that Fig. 5 is an important but visually complex plot, and that a separation of the regimes substantially improves its readability. Therefore, we have added a second column with three additional panels in which the mixed-phase haze (MPH) and mixed-phase cloud (MPC) regimes are displayed side by side (panels d–f). Panels (a–c) now show the ice, aerosol, and liquid regimes. This new layout allows for a direct comparison between MPH and MPC while maintaining the possibility to contrast them with the other cloud regimes.

Changes is the manuscript (Line 249):
The meteorological parameters $T$, RHw, and RHice measured within the different cloud regimes are shown in Fig. 5. The regimes ice, liquid, and aerosol are presented in panels (a–c), while the corresponding distributions for the MPH and MPC regimes are shown in panels (d–f).

Revised figure caption (Fig. 5):
Normalized frequency distribution of the different cloud regimes as a function of environmental conditions such as temperature (a, d), relative humidity over water (b, e) and relative humidity over ice (c, f).

**6: Clarify Novelty and Contribution in Introduction**

Please clearly state the novelty of this study in the Introduction, ideally around Line 50. For example: "In this study, we conduct a detailed investigation of a previously unclassified cloud regime, which we refer to as the mixed-phase haze (MPH)." Is this

the first study to define MPH as a unique regime? Does the novelty stem from high-resolution in-situ observations? How does this work advance beyond previous research? These elements are hinted at throughout the Introduction, but an explicit statement would help readers better understand the contribution.

Reply:

We have adopted this suggestion. We have replaced the sentence in line 50 with the following:

"In this study, thin cloud layers are explicitly included in the analysis. Their occurrence frequency is found to be remarkably high, which motivates the definition of a new cloud regime. Based on microphysical properties measured with high resolution in-situ cloud instruments, this regime is referred to as mixed-phase haze. It is characterized by relatively low particle number concentrations compared to classic mixed-phase clouds. The detailed investigation and characterization of this mixed-phase haze regime form the main focus and novelty of this study."

**7: Line 340–341: Secondary Ice Production?**

The statement that NINP is lower than the haze droplet number (Line 340) might also suggest the influence of secondary ice production. If so, the "Therefore" at Line 341 feels misleading. Please clarify the logical flow here.

Reply:

We greatly appreciate the reviewer's thoughtful consideration that remnants of secondary ice production (SIP) could contribute to the haze particles number concentration observed in the MPH regime. Based on our data, we cannot entirely rule out this possibility, and it is conceivable that a small fraction of particles within the MPH may originate from SIP. However, this fraction is likely minor, since particles generated by SIP are typically expected to exhibit irregular or non-spherical shapes. In contrast, the optical properties measured by the Polar Nephelometer show angular scattering patterns consistent with predominantly spherical particles. We therefore conclude that while a minor contribution of SIP-origin ice particles to the haze number concentration cannot be excluded, the population is clearly dominated by liquid or near-spherical particles.

To acknowledge the possibility of SIP and to phrase our conclusion more cautiously, we have revised Line 341 in the manuscript as follows:

"We therefore consider it most likely that the contribution of ice particles to the haze droplet number concentration is negligible."

Minor comments
- Line 6: "The particle number concentration" — specify what kind of particles (e.g., hydrometeors?).
  We have changed to (line 6)"number concentration of cloud particles"

- Line 17–24: Consider shortening this part of the Introduction. The discussion begins focusing on clouds at Line 25, so the earlier text may be unnecessarily long.

  We thank the reviewer for this suggestion and understand the concern. However, we decided to keep this introductory part unchanged. The in-situ dataset used in this study was collected within the framework of the (AC)³ project (ArctiC Amplification: Climate Relevant Atmospheric and SurfaCe Processes, and Feedback Mechanisms; https://ac3-tr.de/ ), and this introductory section serves to provide the scientific context of the study within the overall project objectives. Therefore, we consider this background information essential for the motivation of our work.

- Line 49: Add citation for the Wegener–Bergeron–Findeisen (WBF) process.
  The original works by Wegener (1912), Bergeron (1935), and Findeisen (1938) describe a physical mechanism of precipitation formation. In cloud physics, however, the term WBF-process is commonly used in a more specific sense to denote the rapid growth of ice crystals at the expense of surrounding supercooled droplets due to differences in saturation vapor pressure over ice and water. To provide concise and accessible references for this commonly used concept, we added the following citations in the revised manuscript:
  "They have hypothesised that these clouds may have been formed by the drying of mixed-phase clouds via the Wegener-Bergeron-Findeisen process (Pruppacher and Klett, 2010; Storelvmo and Tan, 2015)."

- Line 63: "Three research aircrafts" → should be "three research aircraft".
  The word aircraft is already in its correct plural form, so no modification was needed.

- Abstract: Consider combining the two paragraphs into one for better flow.
  This change has been adopted in the revised version.

- Lines 90–100: The description of instruments could benefit from a summary chart. This could include size ranges, acronyms, uncertainties, transmission efficiency, and lower/upper limits.
  In this study, we intentionally kept the instrument description concise, as the same measurement systems and processing methods were already described in detail in Moser et al. (2023b) and Mech et al. (2022). Additional information specific to the HALO-(AC)³ campaign is provided in Ehrlich et al. (2025), to which we refer throughout the manuscript.

- Indentation inconsistencies: e.g., Lines 280–281. Please ensure consistent formatting throughout the manuscript.

  We thank the reviewer for this remark. The formatting issue resulted from different methods used for line breaks in LaTeX. This has now been unified to ensure consistency throughout the manuscript.

References:

Moser, M., Voigt, C., Jurkat-Witschas, T., Hahn, V., Mioche, G., Jourdan, O., Dupuy, R., Gourbeyre, C., Schwarzenboeck, A., Lucke, J., Boose, Y., Mech, M., Borrmann, S., Ehrlich, A., Herber, A., Lüpkes, C., and Wendisch, M.: Microphysical and thermodynamic phase analyses of Arctic low-level clouds measured above the sea ice and the open ocean in spring and summer, Atmospheric Chemistry and Physics, 23, 7257–7280, https://doi.org/10.5194/acp-23-7257-2023, 2023b.

Moser, M.: Microphysical properties and thermodynamic phase of Arctic low-level clouds from in-situ aircraft measurements, PhD thesis, Johannes Gutenberg University Mainz, Mainz, Germany, https://doi.org/10.25358/openscience-11192, 2024

Wegener, A. (1912). "Thermodynamik der Atmosphäre". In: Nature 90.2237, pp. 31–31. DOI:10.1038/090031a0

Bergeron, T. (1935). "On the physics of clouds and precipitation." In: International Union of Geodesy and Geophysics.

Findeisen, F. (1938). "Kolloid-meteorologische Vorgänge bei Niederschlagsbildung". In: Meteor.

Pruppacher, H. and Klett, J.: Microphysics of Clouds and Precipitation, Atmospheric and Oceanographic Sciences Library, Springer Netherlands, ISBN 9780306481000, https://doi.org/10.1007/978-0-306-48100-0, 2010.

Storelvmo, T. and Tan, I.: The Wegener-Bergeron-Findeisen process – Its discovery and vital importance for weather and climate, Meteorologische Zeitschrift, 24, 455–461, https://doi.org/10.1127/metz/2015/0626, 2015.

Ehrlich, A., Crewell, S., Herber, A., Klingebiel, M., Lüpkes, C., Mech, M., Becker, S., Borrmann, S., Bozem, H., Buschmann, M., Clemen, H.-C., De La Torre Castro, E., Dorff, H., Dupuy, R., Eppers, O., Ewald, F., George, G., Giez, A., Grawe, S., Gourbeyre, C., Hartmann, J., Jäkel, E., Joppe, P., Jourdan, O., Jurányi, Z., Kirbus, B., Lucke, J., Luebke, A. E., Maahn, M., Maherndl, N., Mallaun, C., Mayer, J., Mertes, S., Mioche, G., Moser, M., Müller, H., Pörtge, V., Risse, N., Roberts, G., Rosenburg, S., Röttenbacher, J., Schäfer, M., Schaefer, J., Schäfler, A., Schirmacher, I., Schneider, J., Schnitt, S., Stratmann, F., Tatzelt, C., Voigt, C., Walbröl, A., Weber, A., Wetzel, B., Wirth, M., and Wendisch, M.: A comprehensive in situ and remote sensing data set collected during the HALO-(AC)[3] aircraft campaign, Earth System Science Data, 17, 1295–1328, https://doi.org/10.5194/essd-17-1295-2025, 2025

---

## Author Comment (AC3)

Review of *"The Arctic Low-Level Mixed-Phase Haze Regime and its Microphysical Differences to Mixed-Phase Clouds"*, by Manuel Moser, Christiane Voigt, Oliver Eppers, Johannes Lucke, Elena De La Torre Castro, Johanna Mayer, Regis Dupuy, Guillaume Mioche, Olivier Jourdan, Hans-Christian Clemen, Johannes Schneider, Philipp Joppe, Stephan Mertes, Bruno Wetzel, Stephan Borrmann, Marcus Klingebiel, Mario Mech, Christof Lüpkes, Susanne Crewell, André Ehrlich, Andreas Herber, and Manfred Wendisch, egusphere-2025-3876.

**Response to the Community Comment**

Dear Martina Krämer,

We sincerely thank you for your detailed and constructive community comment. We highly appreciate the careful comparison you provided between the PSDs observed in Costa-17 and those presented in Moser-25, as well as the broader context you offer for interpreting our results.
In the following, we address each of the recommendations. Our replies are highlighted in blue, and the corresponding revisions implemented in the manuscript are highlighted in orange.

**Recommendations**
**1.** To contextualize the study within the broader landscape of existing research, I recommend to include the comparison with Costa-17 into the manuscript, in particular to point out:

- a: the similarity of the PSDs found by Moser-25 and Costa-17

  Reply: We thank the CC1 for highlighting the direct comparison between the PSDs of the MPH and MPC regimes presented in Moser-25 and the mixed-phase cloud types reported in Costa-17. This comparison indeed shows a very good agreement between the respective PSD shapes. We fully acknowledge this point and now explicitly mention in the manuscript that the PSDs align closely with those identified by Costa-17. Following was added in line 229 to the manuscript:

  The PSDs of both the MPH and MPC regimes agree remarkably well with the two cloud types in the mixed-phase temperature regime identified by Costa et al. (2017). In particular, the PSD of the MPC regime closely resembles the "Type 1" or "coexistence" category of Costa et al. (2017), whereas the PSD of the MPH regime shows strong similarity to their "Type 2" or "large-ice/WBF" category. The Costa et al. (2017) dataset was obtained with comparable in-situ cloud instrumentation but includes a much broader range of meteorological and geographical conditions, including mid-latitude and tropical mixed-phase clouds as well as clouds outside the atmospheric boundary layer. In contrast, the microphysical interpretation of the small-particle mode in the MPH regime presented in this study is based exclusively on Arctic low-level measurements and therefore explains only a subset of the small particles observed in the Type 2 clouds of Costa et al. (2017).

- a1: indicating that the structures of the PSDs depend not on geography but on temperature and environmental conditions
  Reply: We agree that a key strength of Costa-17 lies in the large variety of environmental conditions sampled, covering a broad temperature spectrum and multiple latitudes. This supports the general notion that PSD shapes may be similar across different regions.
  However, we decided not to emphasize this point in the manuscript, as it could be misleading in the context of our study. A similarity in PSD shape does not necessarily imply similarity in microphysical composition. In contrast to Costa-17, we explicitly show that the small particle mode in the MPH regime consists of wet sea salt aerosol particles originating most likely from the ocean and sea ice surface. Clouds in other regions and situations than Arctic ABL in spring, are influenced by different aerosol sources, which are also represented in the Costa-17 dataset.
  It is therefore plausible that the "Type 2" category in Costa-17 includes multiple microphysical subtypes, where the small particles could include wet sea salt aerosol but also other wet aerosol types or small ice crystals (<20 μm).
  In our study, we specifically analyze only one case: Type 2 mixed-phase clouds within the Arctic boundary layer during spring conditions, for which we can characterize the small particle mode in detail.

- b: that the finding that the small mode particles (< 6 μm) are dissolved sea salt particles is confirmed by the asphericity measurements of Costa-17, and to discuss that Costa-17 found small aspherical particles up to ~20 μm.
  Reply: We appreciate the suggestion to relate our findings to Costa-17 regarding the dissolved sea salt particles. However, we note that Costa-17 does not provide asphericity information for the size range relevant to the haze particles in the MPH regime. The asphericity analysis in Costa-17 is limited to particles between 20 and 50 μm, whereas the haze particles identified in our study are substantially smaller (< 10 μm). Costa-17 reports that particles in the 20–50 μm range are often aspherical. This is fully consistent with our understanding that small ice crystals can occur at sizes below 50 μm which could have influenced these measurements. In contrast, our measurements using the Polar Nephelometer (see Moser et al., 2023) show that the small mode particles in the MPH regime are spherical. Combined with the OPC and ALABAMA measurements, this allows us to identify them as wet sea salt aerosol.

**2.** I also recommend reconsidering the names of the mixed-phase cloud types.

Often, the -well known- two types of mixed-phase clouds are summarized under the acronym MPC. In Moser-25, MPC only includes the cloud type in which small liquid droplets coexist with large ice crystals.

For the completely glaciated mixed-phase clouds the, term MPH (mixed-phase haze) is introduced. This is somewhat misleading because 'haze' (in particular Arctic haze) refers commonly to a size spectrum of grown aerosol particles, not to a cloud. However, this type of completely glaciated cloud is usually considered as glaciated mixed-phase cloud with a very

low concentration of large ice crystals, which, however, represent the dominant mass mode (see Figure 1, bottom panel).

Therefore, I would recommend names that make it clear that both types are clouds, maybe MPC coex for the Type 1 and MPC haze for the second?

Reply: We are thankful for this suggestion and agree that our terminology differs from that used in Costa-17, although the cloud types to which these terms refer partially overlap. The full term "MPH" refers to **Arctic low-level mixed-phase haze**, and this name reflects the conditions under which this cloud type is observed and the microphysical properties. The regime is measured exclusively in the Arctic, which motivates the term "Arctic," and it occurs only within the boundary layer, which is why we use "low-level." Microphysically, the cloud consists of a mixture of large ice crystals and smaller particles, similar to a classic mixed-phase cloud. However, unlike in a classic mixed-phase cloud, the small particles are unactivated cloud droplets and are commonly referred to as haze droplets.
We also note that the term "Arctic haze" is not related to our definition of MPH, which is explicitly clarified in the manuscript (line 343).
In order to connect the terminology used in Costa-17 and Moser-25, we now explicitly use the terms "Type 1; coexistence" and "Type 2; large ice/WBF" in the PSD discussion of the revised manuscript.

References:

Moser, M., Voigt, C., Jurkat-Witschas, T., Hahn, V., Mioche, G., Jourdan, O., Dupuy, R., Gourbeyre, C., Schwarzenboeck, A., Lucke, J., Boose, Y., Mech, M., Borrmann, S., Ehrlich, A., Herber, A., Lüpkes, C., and Wendisch, M.: Microphysical and thermodynamic phase analyses of Arctic low-level clouds measured above the sea ice and the open ocean in spring and summer, Atmospheric Chemistry and Physics, 23, 7257–7280, https://doi.org/10.5194/acp-23-7257-2023, 2023b.

---

## Referee Report (RR1)

Review of manuscript "The Arctic Low-Level Mixed-Phase Haze Regime and its Microphysical Differences to Mixed-Phase Clouds"

The primary change of this manuscript includes the adding in of clustering justification of Table 1, and the updates of Figure 2 and Figure 5. These updates 1) connect the previously published journal paper and the current manuscript, 2) improve the visualization of the mixed phase haze microphysics (Fig.5) over the arctic, and 3) show more information of the sonde data distribution. So far the constraint parts are, 1) the explanation of Fig 6, and the scheme figure Fig 11. All the other sections are logically strong and scientifically reasonable in my opinion. I therefore recommend "technical corrections" in this round of review.

1. Figure 11 was designed to visualize the microphysics of MPC and MPH over the open water and sea ice. I wonder if the MIZ should also be shown in this figure between the sea ice and open water considering Fig 9 and 3 have clusters of sea ice, ocean/land and MIZ. This can be a minor change and mainly rely on Fig 9. Also, on the right edge of this figure, there is only MPH existing over the open water. Were the authors implying anything by doing this on purpose? I would appreciate a brief description of this phenomenon and the motivation in the description.

2. This has been mentioned in the previous round before but the authors didn't do an excellent job in modifying to help the readers' interpretation. Confusion arose when I first read this section about Fig 6 and I am confused again in this round of reading. For this figure to really talk about the aerosols and their size distribution, a background introduction of the measurements/physical principle or an intro level discussion of "Particle Fraction" and "Number of Spectra" would be meaningful. I understand the instrumentation section introduced the technical details of ALABAMA but for the readers' convenience, it is important to stress some easy physics here.

   I appreciate your add in of "Based on this hygroscopic growth factor and the observed haze droplet sizes ranging from 3 µm to 6µm, the estimated dry diameters of the solute particles are between 1µm and 3µm, calculated from the observed wet diameter range (3–6µm) divided by the 25th and 75th percentile of the hygroscopic growth factor." Actually, I personally think the details (e.g. reasons of few particles were analyzed by the ALABAMA above 3µm) are well explained. But not the physics behind the figure, e.g. "Nevertheless, the domination of SSA at larger particle sizes and the potential of hygroscopic growth for dry NaCl particles to sizes around 5µm, suggest wet SSA contributing to the observed haze particles." Please try adjusting the language to be considerate for the readers in your final draft since Figure 6 so far serves as the least well-explained figure in this outstanding quality manuscript.